# Score Distillation via Reparametrized DDIM

Artem Lukoianov[1]      Haitz Sáez de Ocáriz Borde[2]

Kristjan Greenewald[3]      Vitor Campagnolo Guizilini[4]      Timur Bagautdinov[5]

Vincent Sitzmann[1]      Justin Solomon[1]

[1]Massachusetts Institute of Technology      [2]University of Oxford      [3]MIT-IBM Watson AI Lab, IBM Research      [4]Toyota Research Institute      [5]Meta Reality Labs Research
https://lukoianov.com/sdi/

## Abstract

While 2D diffusion models generate realistic, high-detail images, 3D shape generation methods like Score Distillation Sampling (SDS) built on these 2D diffusion models produce cartoon-like, over-smoothed shapes. To help explain this discrepancy, we show that the image guidance used in Score Distillation can be understood as the velocity field of a 2D denoising generative process, up to the choice of a noise term. In particular, after a change of variables, SDS resembles a high-variance version of Denoising Diffusion Implicit Models (DDIM) with a differently-sampled noise term: SDS introduces noise i.i.d. randomly at each step, while DDIM infers it from the previous noise predictions. This excessive variance can lead to over-smoothing and unrealistic outputs. We show that a better noise approximation can be recovered by inverting DDIM in each SDS update step. This modification makes SDS's generative process for 2D images almost identical to DDIM. In 3D, it removes over-smoothing, preserves higher-frequency detail, and brings the generation quality closer to that of 2D samplers. Experimentally, our method achieves better or similar 3D generation quality compared to other state-of-the-art Score Distillation methods, all without training additional neural networks or multi-view supervision, and providing useful insights into relationship between 2D and 3D asset generation with diffusion models.

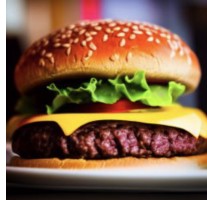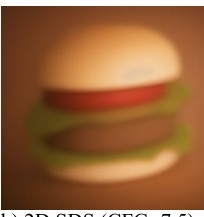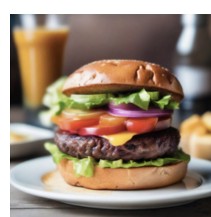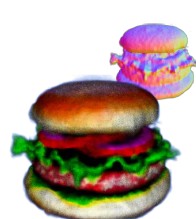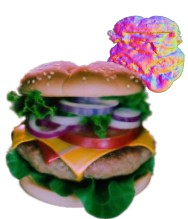

a) DDIM (CFG=7.5)      b) 2D SDS (CFG=7.5)      c) 2D ours (CFG = 7.5)      d) 3D SDS (CFG=100)      e) 3D ours (CFG=7.5)

Figure 1: Score Distillation Sampling (SDS) "distills" 3D shapes from 2D image generative models like DDIM. While DDIM produces high-quality images (a), the same diffusion model, yields blurry results with SDS in the task of 2D image generation (b); in 3D, SDS yields over-saturated and simplified shapes (d). By replacing the noise term in SDS to agree with DDIM, our algorithm better matches the quality of the diffusion model in 2D (c) and significantly improves 3D generation (e).

38th Conference on Neural Information Processing Systems (NeurIPS 2024).

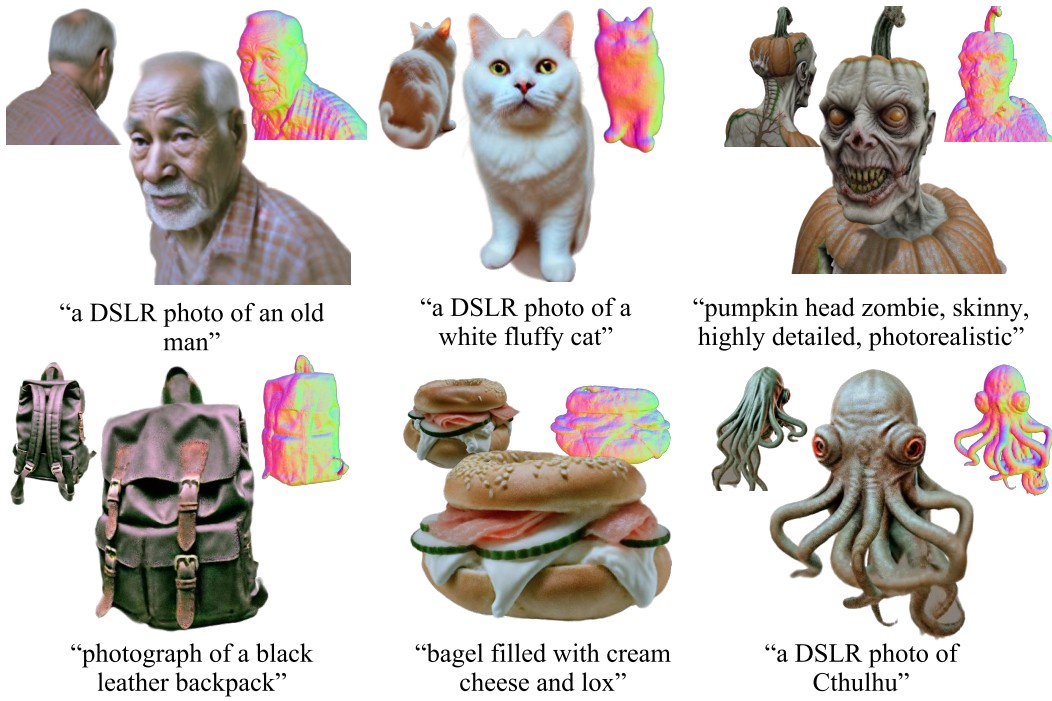

"a DSLR photo of an old man"

"a DSLR photo of a white fluffy cat"

"pumpkin head zombie, skinny, highly detailed, photorealistic"

"photograph of a black leather backpack"

"bagel filled with cream cheese and lox"

"a DSLR photo of Cthulhu"

Figure 2: Examples of 3D objects generated with our method.

# 1 Introduction

Image generative modeling saw drastic quality improvement with the advent of text-to-image diffusion models [1] trained on billion-scale datasets [2] with large parameter counts [3]. From a short prompt, these models generate photorealistic images, with strong zero-shot generalization to new classes [4]. Efficient training methods for image data, combined with Internet-scale datasets, enabled the development of these models. However, applying similar techniques to domains where huge datasets are scarce, such as 3D shape generation, remains challenging.

The need for 3D objects in downstream applications like vision, graphics, and robotics motivated methods like Score Distillation Sampling (SDS) [5] and Score Jacobian Chaining (SJC) [6], which optimize volumetric 3D representations [7, 8, 9] using queries to a 2D generative model [10]. In every iteration, SDS renders the current state of the 3D representation from a random viewpoint, adds noise to the result, and then denoises it using the pre-trained 2D diffusion model conditioned on a text prompt. The difference between the added and predicted noise is used as a gradient-style update on the rendered images, which is propagated to the parameters of the 3D model. The underlying 3D representation helps make the generated images multi-view consistent, and the 2D model guides individual views towards a learned distribution of realistic images.

In practice, however, as noted in [11, 12, 13], SDS often produces 3D representations with over-saturated colors and over-smoothed textures (fig. 1d), not matching the quality of the underlying 2D model. Existing approaches tackling this problem improve quality at the cost of expensive re-training or fine-tuning of the image diffusion model [11], complex multi-stage handling of 3D representations like mesh extraction and texture fine-tuning [14, 15, 11], or altering the SDS guidance [13, 12, 16].

As an alternative to engineering-based improvements to SDS, in this paper we reanalyze the vanilla SDS algorithm to understand the underlying source of artifacts. Our key insight is that the SDS update rule steps along an approximation of the DDIM velocity field. In particular, we derive Score Distillation from DDIM with a change of variables to the space of single-step denoised images. In this light, SDS updates are nearly identical to DDIM updates, apart from one difference: while DDIM samples noise *conditionally* on the previous predictions, SDS resamples noise i.i.d. in every iteration. This breaks the denoising trajectory for each independent view and introduces excessive variance. Our perspective unifies DDIM and SDS and helps explain why SDS can produce blurry and

over-saturated results: the variance-boosting effect of noisy guidance is usually mitigated with high Classifier-Free Guidance (CFG) [17] to reduce sample diversity at the cost of over-saturation [18].

Based on our analysis, we propose an alternative score distillation algorithm dubbed Score Distillation via Inversion (SDI), closing the gap to DDIM. We obtain the conditional noise required for consistency of the denoising trajectories by inverting DDIM on each step of score distillation (fig. 5). This modification yields 3D objects with high-quality textures consistent with the 2D diffusion model (fig. 1e). Moreover, in 2D, our method closely approximates DDIM while preserving the incremental generation schedule of SDS (fig. 1c).

Our key contributions are as follows:

- We prove that guidance for each view in the SDS algorithm is a simplified reparameterization of DDIM sampling: vanilla SDS samples random noise at each step, while DDIM keeps the trajectories consistent with previously-predicted noise.
- We propose a new method titled Score Distillation via Inversion (SDI), which replaces the problematic random noise sampling in SDS with prompt-conditioned DDIM inversion and significantly improves 3D generation, closing the quality gap to samples from the 2D model.
- We systematically compare SDI with the state-of-the-art Score Distillation algorithms and show that SDI achieves similar or better generation quality, while not requiring training additional neural networks or multiple generation stages.

## 2    Related work

**3D generation by training on multi-view data.** Recent 3D generation methods leverage multi-view or 3D data. Zero123 [19] and MVDream [20] generate consistent multi-view images from text; a 3D radiance field is then obtained via score distillation. Video generative models can be fine-tuned on videos of camera tracks around 3D objects, similarly yielding a model that samples multi-view consistent images to train a 3D radiance field [21, 22]. Diffusion with Forward Models [23] and Viewset Diffusion [24] directly train 3D generative models from 2D images. While these methods excel at generating multi-view consistent, plausible 3D objects, they depend on multi-view data with known camera trajectories, limiting them to synthetic or small bundle-adjusted 3D datasets. We instead focus on methods that require only single-view training images.

**Distilling 2D into 3D.** Score Distillation was introduced concurrently in Dreamfusion or SDS [5], Score Jacobian Chaining (SJC) [6], and Magic3D [14]. The key idea is to use a frozen diffusion model trained on 2D images and "distill" it into 3D assets. A volumetric representation of the shape is rendered from a random view, perturbed with random noise, and denoised with the diffusion model; the difference between added and predicted noise is used to improve the rendering. These works, however, suffer from over-smoothing and lack of detail. Usually a high value of classifier free guidance (CFG ≈ 100) [17] is used to reduce variance at the cost of over-saturation [18].

ProlificDreamer [11] generates sharp, detailed results with standard CFG values (≈ 7.5) and without over-saturation. The key idea is to overfit an additional diffusion model to specifically denoise the current 3D shape estimate. Fine-tuning the second model, however, is cumbersome and theoretical justification for this change is still unclear. Recent papers further improve on ProlificDreamer's results or try to explain its behavior. SteinDreamer [25], for example, hypothesizes that ProlificDreamer's improvements come from variance reduction of the sampling procedure [26].

Other papers propose heuristics that improve SDS. [12, 16, 27] decompose the guidance terms and speculate about their relative importance. Empirically, visual quality can be improved by suppressing the denoising term with negative prompts [12] or highlighting the classification term [16]. [14, 15, 11, 28] use multi-stage optimization: they first train a volumetric representation and then extract a mesh or voxel grid to fine-tune geometry and texture. HiFA [13] combines ad-hoc techniques: additional supervision in latent space, time annealing, kernel smoothing, $z$-variance regularization, and a pretrained mono-depth estimation network to improve the quality of single-stage NeRF generation.

LucidDreamer, or Interval Score Matching (ISM), hypothesizes based on empirical evidence that in SDS, the high-variance random noise term and large reconstruction error of a single-step prediction togehter cause over-smoothing [29]. Based on these observations, the authors replace the random noise term in SDS with noise obtained by running DDIM inversion and introduce multi-step denoising to improve reconstruction quality. As we will show in section 4, the update rule of ISM can be seen

**Prompt** – "colored photograph of an old man"

**CFG Value**       1.          5.        10.        30.        100.

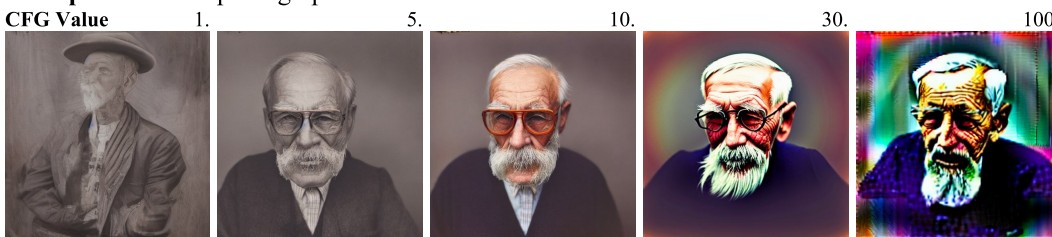

Figure 3: The effect of CFG values on 2D generation with StableDiffusion 2.1 [1]. For small values, the model tends to ignore certain words in the prompt. For high values, images become over-saturated.

as a special case of our formulation. Moreover, our analysis reveals that the added noise should be inferred *conditionally* on the text prompt $y$, which further improves quality.

In this work, rather than augmenting the SDS pipeline or relying on heuristics, we derive Score Distillation through the denoising process of DDIM and propose a simple modification of SDS that significantly improves 3D generation.

## 3 Background

**Diffusion models.** Denoising Diffusion Implicit Models (DDIM) generate images by reversing a diffusion process [30, 31, 32]. After training a denoiser $\epsilon_\theta^t$ and freezing its weights $\theta$, the denoising process can be seen as an ODE on rescaled noisy images $\bar{x}(t) = x(t)/\sqrt{\alpha(t)}$. Given a prompt $y$ and current time step $t \in [0, 1]$, the denoising process satisfies:

$$\frac{d\bar{x}(t)}{dt} = \epsilon_\theta^t\big(\sqrt{\alpha(t)}\bar{x}(t), y\big)\frac{d\sigma(t)}{dt}, \tag{1}$$

where $\bar{x}(1)$ is sampled from a Gaussian distribution, $\sigma(t) = \sqrt{1 - \alpha(t)}/\sqrt{\alpha(t)}$, and $\alpha(t)$ are scaling factors. When discretized with forward Euler, this equation yields the following update to transition from step $t$ to a less noisy step $t - \tau < t$:

$$\bar{x}(t - \tau) = \bar{x}(t) + \epsilon_\theta^t\big(\sqrt{\alpha(t)}\bar{x}(t), y\big)\left[\sigma(t - \tau) - \sigma(t)\right]. \tag{2}$$

The DDIM ODE can also be integrated in reverse direction to estimate $\bar{x}(t)$ for any $t \in [0, 1]$ from a clean image $x_0$. This operation is called *DDIM inversion* and is studied in multiple works [33, 34].

**Classifier-free guidance.** Classifier-free guidance (CFG) [17] provides high-quality conditional samples without gradients from auxilary models [35]. CFG modifies the noise prediction $\hat{\epsilon}_\theta^t$ (score function) by linearly combining conditional and unconditional predictions:

$$\epsilon_\theta^t\big(x(t), y\big) = \hat{\epsilon}_\theta^t(x(t), \varnothing) + \gamma \cdot \big(\hat{\epsilon}_\theta^t(x(t), y) - \hat{\epsilon}_\theta^t(x(t), \varnothing)\big), \tag{3}$$

where the guidance scale $\gamma$ is a scalar, with $\gamma = 0$ corresponding to unconditional sampling and $\gamma = 1$ to conditional. In practice, larger values $\gamma > 1$ are necessary to obtain high-quality samples at a cost of reduced diversity and extreme over-saturation. We demonstrate the effect of different CFG values in fig. 3. For the rest of the paper we use the modified CFG version of the denoiser $\epsilon_\theta^t\big(x(t), y\big)$.

**Score Distillation.** Diffusion models efficiently generate images and can learn to represent common objects from arbitrary angles [36] and with varying lighting [5]. Capitalizing on this success, Score Distillation Sampling (SDS) [5] distills a pre-trained diffusion model $\epsilon_\theta^t$ to produce a 3D asset. In practice, the 3D shape is usually parameterized by a NeRF [7], InstantNGP [8], or Gaussian Splatting [9]. Multiple works additionally extract an explicit representation for further optimization [14, 15, 11, 28]. We use InstantNGP [8] to balance between speed and ease of optimization.

Denote the parameters of a differentiable 3D shape representation by $\psi \in \mathbb{R}^d$ and differentiable rendering by a function $g(\psi, c) : \mathbb{R}^d \times C \to \mathbb{R}^{N \times N}$ that returns an image given camera parameters $c \in C$. Intuitively, in each iteration, SDS samples $c$, renders the corresponding (image) view $g(\psi, c)$, perturbs it with $\epsilon \sim \mathcal{N}(0, I)$ to level $t \sim [0, 1]$, and denoises it with $\epsilon_\theta^t$; the difference between the true and predicted noise is propagated to the parameters of the 3D shape. More formally, after sampling the camera view $c$ and randomly drawing a time $t$, SDS renders the volume and adds

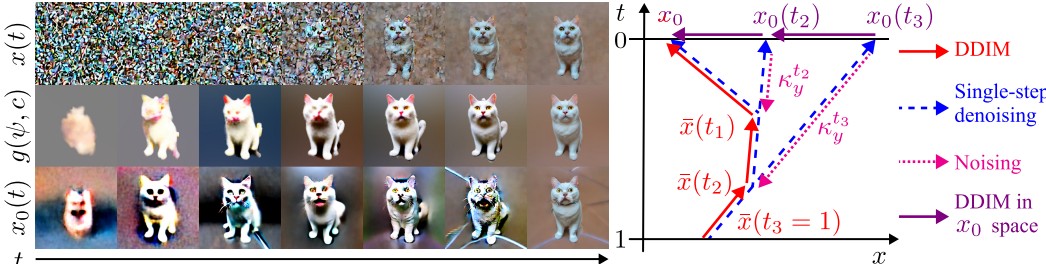

Figure 4: Left: Evolution of variables in Score Distillation with time. The top row depicts how noisy images $x(t)$ evolve during 2D generation; the middle row shows evolution of a NeRF for 3D generation; and the bottom row shows how the single step denoised variable $x_0(t)$ changes with $t$. Right: Each step of DDIM steps toward a denoised image. This can be seen as a step to $x_0(t)$ and a step back to a slightly less noisy image. Through a change-of-variables we obtain a process on $x_0(t)$.

Gaussian noise $\epsilon$ to obtain a noisy image $x(t) = \sqrt{\alpha(t)}g(\psi, c) + \sqrt{1 - \alpha(t)}\epsilon$. Then, SDS improves the generated volume by using a gradient(-like) direction to update its parameters $\psi$:

$$\nabla_\psi \mathcal{L}_{SDS} = \mathbb{E}_{t,\epsilon,c}\sigma(t)\left[\epsilon_\theta^t\big(x(t), y\big) - \epsilon\right]\frac{\partial g}{\partial \psi}. \tag{4}$$

We refer to the term $\left[\epsilon_\theta^t\big(x(t), y\big) - \epsilon\right]$ as *guidance* in score distillation, as it 'guides' the views of the shape. In theory, this expression may not correspond to the true gradient of a function and there are many hypotheses about its effectiveness [5, 25, 12, 16, 26]. In this work, we show that instead it can be seen as a high-variance version of DDIM after a change of variables.

## 4   Linking SDS to DDIM

**Discrepancy in image sampling.** Beyond the lack of formal justification of eq. (4), in practice SDS results are over-saturated and miss details for high CFG values, while they are blurry for low CFG values. To illustrate this phenomenon, fig. 1 shows a simple experiment, inspired by [11]: We replace the volumetric representation in eq. (4) with an image $g_{2D}(\psi_{2D}, c) := \psi_{2D} \in \mathbb{R}^{N \times N}$. In this case, SDS becomes an image generation algorithm that can be compared to other sampling algorithms like DDIM [37]. Even in this 2D setting, SDS fails to generate sharp details, while DDIM with the same underlying diffusion model produces photorealistic results, motivating our derivation below.

**Why not use DDIM as guidance?** Given the experiment above, a natural question to ask is if it is possible to directly use DDIM's update direction from eq. (1) as SDS guidance in eq. (4) to update the 3D representation. The problem with this approach lies in the discrepancy between the training data of the denoising model and the images generated by rendering the current 3D representation. More specifically, the denoising network expects an image with a certain level of noise corresponding to time $t$ as defined by the forward (noising) diffusion process, whereas renderings of 3D representations $g(\psi, c)$ evolve from a blurry cloud to a well-defined sample (fig. 4 left).

**Evolution of $x_0(t)$.** Instead of seeing DDIM as a denoising process defined on the space of noisy images $x(t)$, we reparametrize it to a new variable:

$$x_0(t) \triangleq \bar{x}(t) - \sigma(t)\epsilon_\theta^t\big(x(t), y\big). \tag{5}$$

In words, $x_0(t)$ is the noisy image at time $t$ denoised with a single step of noise prediction. Empirically, the evolution of $x_0(t)$ is similar to the evolution of $g(\psi, c)$—from blurry to sharp. The left side of fig. 4 compares these processes. This similarity motivates us to rewrite eq. (1) in terms of $x_0(t)$, and to understand SDS as applying similar updates to the renderings of its 3D representation.

**Reparametrizing DDIM.** Figure 4 (right) shows schematically how the DDIM update to $\bar{x}(t)$ alternates between denoising to obtain $x_0(t)$ and adding the predicted noise back to get a cleaner $\bar{x}(t - \tau)$. Based on the intuition above, we reorder the steps, adding noise to $x_0(t)$ and then denoising to estimate $x_0(t - \tau)$. Consider neighboring time points $t$ and $t - \tau < t$ in discretized DDIM eq. (2) (lower time means less noise). We rewrite eq. (2) using the definition of $x_0(t)$ from eq. (5) to find

$$x_0(t - \tau) = x_0(t) - \sigma(t - \tau)\left[\epsilon_\theta^{t-\tau}\big(x(t - \tau), y\big) - \epsilon_\theta^t\big(x(t), y\big)\right], \tag{6}$$

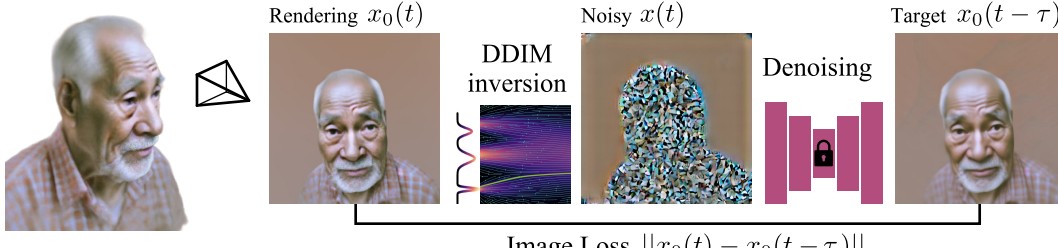

Figure 5: **Overview of SDI.** At each training iteration, SDI renders a random view of the 3D shape, runs DDIM inversion up to the noise level $t$, and denoises the image with a pre-trained diffusion model for noise level $t - \tau$. Finally, the denoised image is back-propagated into the 3D shape.

which is consistent with the intuition behind SDS: improving an image involves perturbing the current image and then denoising it with a better noise estimate. We cannot directly apply eq. (6) to SDS in 3D, since it depends on $x(t)$; if we think of $x_0(t)$ as similar to a rendering of the 3D representation for some camera angle, it is unclear how to obtain a consistent set of preimages $x(t)$ at each step of 3D generation. From eq. (5), however, $x(t)$ should satisfy the following fixed point equation:

$$x(t) = \sqrt{\alpha(t)}x_0(t) + \sqrt{1 - \alpha(t)}\epsilon_\theta^t\big(x(t), y\big), \tag{7}$$

or rewritten in terms of noise $\epsilon = [x(t) - \sqrt{\alpha(t)}x_0(t)]/\sqrt{1 - \alpha(t)}$:

$$\epsilon = \epsilon_\theta^t\big(\sqrt{\alpha(t)}x_0(t) + \sqrt{1 - \alpha(t)}\epsilon, y\big). \tag{8}$$

Define $\kappa_y^t\big(x_0(t)\big) = \epsilon$ as a solution of this equation given $x_0(t)$. Then, we can write:

$$\epsilon_\theta^t\big(x(t), y\big) = \kappa_y^t\big(x_0(t)\big) \quad \text{and} \quad x(t - \tau) = \sqrt{\alpha(t - \tau)}x_0(t) + \sqrt{1 - \alpha(t - \tau)}\kappa_y^t\big(x_0(t)\big). \tag{9}$$

Thus, eq. (6) turns into:

$$x_0(t-\tau) = x_0(t) - \sigma(t-\tau)\Big[\underbrace{\epsilon_\theta^{t-\tau}\big(\overbrace{\sqrt{\alpha(t-\tau)}x_0(t) + \sqrt{1-\alpha(t-\tau)}\kappa_y^t\big(x_0(t)\big)}^{x_0 \text{ noised with } \kappa_y^t \text{ to time } t-\tau}, y\big)}_{\text{predicted noise}} - \underbrace{\kappa_y^t\big(x_0(t)\big)}_{\text{noise sample } \kappa_y^t}\Big]. \tag{10}$$

We can already see that the structure of eq. (10) is very similar to the SDS update rule in eq. (4). Note that the update direction in eq. (10) is the same as in the SDS update rule in eq. (4), where $\kappa_y^t$ plays the role of the random noise sample $\epsilon$. We could use it as a guidance for the 3D generative process in SDS by replacing $\epsilon$ in eq. (4) with $\kappa_y^t(x_0(t))$. In practice, however, it is hard to solve eq. (8), as $\epsilon_\theta^t$ is high-dimensional and nonlinear. In an unconstrained 2D generation, $\kappa_y^t$ can be cached from a previous denoising step, matching the update step to DDIM exactly as in fig. 1c. In 3D, however, this is impossible due to the simultaneous optimization of multiple views and projections to the space of viable 3D shapes. Below we show that a naïve approximation replacing $\kappa_y^t$ with a Gaussian yields SDS, and we will propose alternatives that are more faithful to the derivation above.

**SDS as a special case.** From eq. (10), to get a cleaner image, we need to bring the current image to time $t$ with noise sample $\kappa_y^t$, denoise the obtained image, and then subtract the difference between added and predicted noise from the initial image. A coarse approximation of $\kappa_y^t$ uses i.i.d. random noise $\kappa_{\text{SDS}}(x_0(t)) \sim \mathcal{N}(0, I)$, matching the forward process by which diffusion adds noise. This choice of $\kappa_{\text{SDS}}$ precisely matches the update rule eq. (10) to the SDS guidance in eq. (4).

**ISM as a special case.** The main update formula in Interval Score Matching (ISM) [29] is a particular case of eq. (10), where $\kappa_y^t$ is obtained via DDIM inversion without conditioning on the text prompt $y$. As demonstrated in section 5, DDIM inversion approximates a solution of eq. (10), explaining the improved performance of ISM. However, our analysis suggests that an even better-performing $\kappa_y^t$ should incorporate the text prompt $y$, which further improves the results and avoids over-saturation.

## 5  Score Distillation via Inversion (SDI)

As we have shown, SDS follows the velocity field of reparametrized DDIM in eq. (10), when $\kappa_{\text{SDS}}(x_0(t))$ is randomly sampled in each step. Our derivation, however, suggests that $\kappa_{\text{SDS}}(x_0(t))$

could be improved by bringing it closer to a solution of the fixed-point equation in eq. (8). Indeed, randomly sampling $\kappa_{\text{SDS}}$ as in Dreamfusion yields excessive variance and blurry results for standard CFG values, while using higher CFG values leads to over-saturation and lack of detail. On the other hand, solving eq. (8) exactly is challenging due to its high dimensionality and nonlinearity.

Like ISM [29], we suggest to obtain $\kappa_y^t$ by inverting DDIM, that is, by integrating the ODE in eq. (1) with $t$ evolving backwards (from images to noise) as in [37, 33, 34]. As we can see in eq. (8), $\kappa_y^t$ should be a function of the text prompt $y$, leading us to perform DDIM inversion *conditionally* on $y$, unlike ISM. This process approximates but is not identical to the exact solution for $\kappa_y^t$: fixed points of eq. (8) invert a single large step of DDIM, while running the ODE in reverse inverts the entire DDIM trajectory. We ablate alternative choices for $\kappa_y^t$ in section 6.2 and conclude that in practice DDIM inversion offers the best approximation quality. Additionally, to match the iterative nature of DDIM, we employ a linear annealing schedule of $t$. We refer to our modified version of SDS as *Score Distillation via Inversion*, or *SDI*.

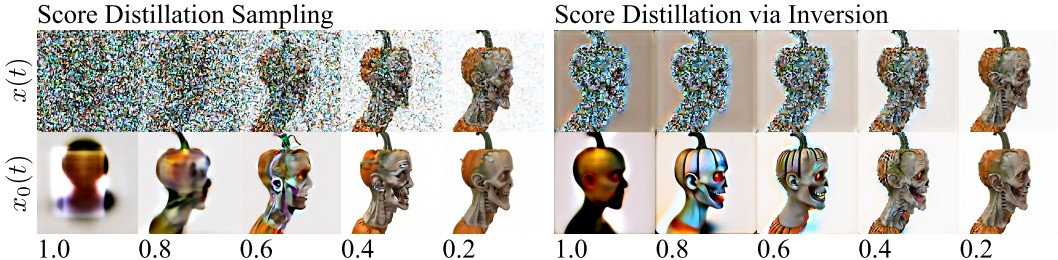

Figure 6: Comparison of intermediate variables in SDS and SDI (ours) for different timesteps $t$. Starting with a rendering of a 3D shape we demonstrate how each algorithm perturbs it ($x(t)$ variable on the top row) and how it is denoised with a single step of diffusion ($x_0(t)$ variable on the bottom row). The prompt used is "Pumpkin head zombie, skinny, highly detailed, photorealistic, side view."

Figure 6 shows the effect of inferring the noise via DDIM inversion instead of sampling it randomly. The special structure of the improved $\kappa_y^t$ results in more consistent single-step generations and produces intricate features at earlier times. When inverted and not sampled, the noise appears 'in the right place': in SDS the noise covers the whole view, including the background, whereas in ours the noise is concentrated on the meaningful part of the 3D shape. This improves geometric and temporal coherency for $x_0(t)$ predictions even for large $t$. The reduced variance drastically increases sharpness and level of detail. Moreover, it allows to reduce CFG value of generation $\gamma_{\text{fwd}}$ to the standard 7.5, avoiding over-saturation. Another interesting finding is that DDIM inversion works best when the reverse integration is performed with negative CFG $\gamma_{\text{inv}} = -\gamma_{\text{fwd}} = -7.5$. The overview of our method is presented in fig. 5, the details about inversion algorithm are presented in section 6.2, and the implementation details are discussed in appendix A.

## 6   Experiments

### 6.1   3D generation

We demonstrate the high-fidelity 3D shapes generated with our algorithm in fig. 2 and provide more examples of 360° views in appendix H.2. A more detailed qualitative and quantitative comparison of our method with ISM [29] is provided in appendix B. Additionally, we report the diversity of the generated shapes in appendix C.

**Qualitative comparisons.** Figure 7 compares 3D generation quality with the results reported in past work using a similar protocol to [11, 12]. For the baselines we chose: Dreamfusion [5] (the work we build on), Noise Free Score Distillation (NFSD) [12] (uses negative prompts in SDS), ProlificDreamer [11] (fine-tunes and trains a neural network to denoise the 3D shape), Interval Score Matching (ISM) [29] (obtain the noise sample by inverting DDIM and perform multi-step denoising), and HiFA [13] (guides in both image and latent spaces, regularizes the NeRF, and supervises the geometry with mono-depth estimation). Our figures indicate that Score Distillation via Inversion (SDI) yields similar or better results compared with state-of-the-art. Appendix H.3 presents more extensive comparison.

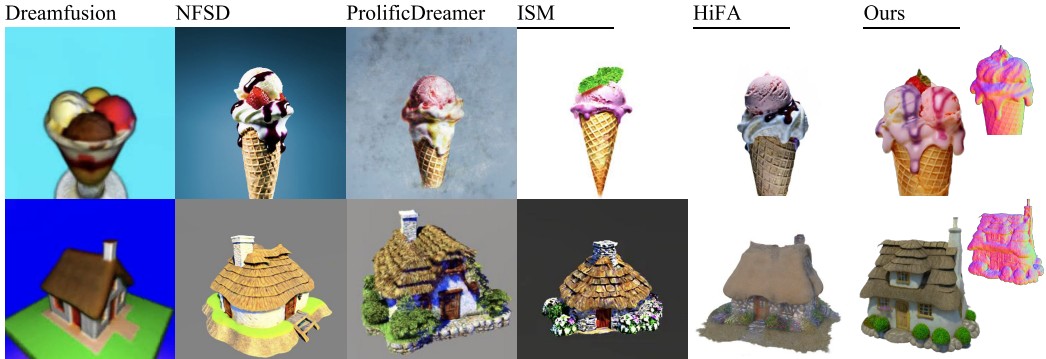

Figure 7: Comparison of 3D generation with other methods using their reported results. The prompts are "An ice cream sundae" and "A 3D model of an adorable cottage with a thatched roof".

**Quantitative comparison.** We follow [5, 16, 25] to quantitatively evaluate generation quality. Table 1 provides CLIP scores [38] to measure prompt-generation alignment, computed with `torchmetrics` [39] and the ViT-B/32 model [40]. We also report ImageReward (IR) [41] to imitate possible human preference. We include CLIP Image Quality Assessment (IQA) [42] to measure quality ("Good photo" vs. "Bad photo"), sharpness ("Sharp photo" vs. "Blurry photo"), and realism ("Real photo" vs. "Abstract photo"). For each method, we test 43 prompts with 50 views. For multi-stage baselines, we run only the first stage for fair comparison. We report the percentage of generations that run out-of-memory or generate an empty volume as diverged ("Div." in the table). as well as mean run time and VRAM usage. For VRAM, we average the maximum usage of GPU memory between runs. As many baselines are not open-source, we use their implementations in `threestudio` [43]. SDI outperforms SDS and matches or outperforms the quality of state-of-the-art methods, offering a simple fix to SDS without additional supervision or multi-stage training.

Table 1: Quantitative comparisons to baselines for text-to-3D generation, evaluated by CLIP Score and CLIP IQA. We report mean and standard deviation across 43 prompts and 50 views for each.

| Method | CLIP Score (↑) | CLIP IQA (%) ↑ | | | IR (↑) | Div. (%) ↓ | Time | VRAM |
|---|---|---|---|---|---|---|---|---|
| | | "quality" | "sharpness" | "real" | | | | |
| SDS [5], $10k$ steps | $29.81 \pm 2.49$ | $76 \pm 6.6$ | $99 \pm 1.2$ | $98 \pm 2.4$ | $-1.51 \pm 0.83$ | 18.6 | 66min | 6.2GB |
| SJC [6], $10k$ steps | $30.39 \pm 1.98$ | $76 \pm 6.4$ | $99 \pm 0.1$ | $98 \pm 1.1$ | $-1.76 \pm 0.51$ | 11.6 | 13min | 13.1GB |
| VSD [11], $25k$ steps | $33.31 \pm 2.39$ | $77 \pm 6.7$ | $98 \pm 1.3$ | $96 \pm 4.4$ | $-1.17 \pm 0.58$ | 23.2 | 334min | 47.9GB |
| ESD [44], $25k$ steps | $32.79 \pm 2.15$ | $77 \pm 7.2$ | $98 \pm 1.2$ | $97 \pm 2.7$ | $-1.20 \pm 0.64$ | 14.0 | 331min | 46.8GB |
| HIFA [13], $25k$ steps | $32.80 \pm 2.35$ | $81 \pm 6.5$ | $98 \pm 1.5$ | $97 \pm 1.2$ | $-1.16 \pm 0.69$ | 4.7 | 235min | 46.4GB |
| **SDI(ours),** $10k$ **steps** | $33.47 \pm 2.49$ | $82 \pm 6.3$ | $98 \pm 1.3$ | $97 \pm 1.2$ | $-1.18 \pm 0.59$ | 4.7 | 119min | 39.2GB |

## 6.2 Ablations

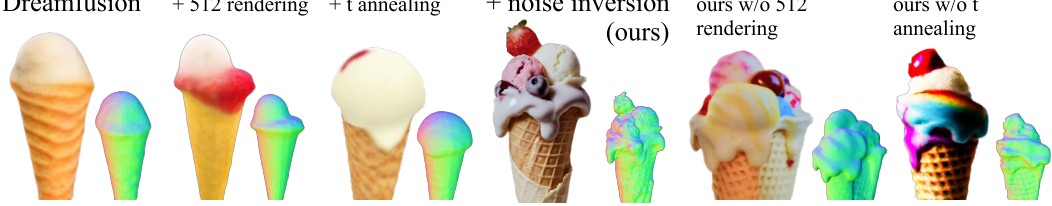

Figure 8: Ablation study of proposed improvements.

**Proposed improvements.** Figure 8 ablates the changes we implement on top of SDS. Starting from Dreamfusion [5] with CFG 7.5 we incrementally add: higher NeRF rendering resolution ($64 \times 64$ to $512 \times 512$), linear schedule on $t$, and—our core contribution—DDIM inversion. The results clearly demonstrate that the main improvement in quality comes from the inferred noise.

**Choice of $\kappa_y^t(x)$.** The key component of our algorithm is the choice of inferred noise $\kappa_y^t(x)$. In theory, $\kappa_y^t(x)$ should solve eq. (8), but it is impractical to do so. Hence, in fig. 9 we compare the following choices of $\kappa_y^t(x)$ and report their numerical errors:

- *Random, resampled:* Sample $\kappa_y^t(x)$ in each new update step from $\mathcal{N}(0, I)$ ;
- *Random, fixed:* Sample $\kappa_y^t(x)$ from $\mathcal{N}(0, I)$ once, and keep it fixed for each iteration;
- *Fixed point iteration:* Since the optimal solution is a fixed point of eq. (8), initialize $\kappa_y^t(x) \sim \mathcal{N}(0, I)$ and run fixed point iteration [45] for 10 steps (in practice, more steps did not help).
- *SGD optimization:* Optimize $\kappa_y^t(x)$ via gradient decent for 10 steps, initializing with noise.
- *DDIM inversion:* Run DDIM inversion for $int(10t)$ (fewer steps for smaller $t$) steps to time $t$, with negative CFG $\gamma_{\text{inv}} = -7.5$ for inversion and positive $\gamma_{\text{fwd}} = 7.5$ for forward inference.

The left side of fig. 9 compares the choices for 3D generation qualitatively; the right side plots error induced in eq. (8) (rescaled to $x_0$ variable due to its ambiguity around 0). As can be seen, the regressed noise has a big impact on the final generations. Both fixed point iteration and optimization via gradient descent fail to improve the approximation of $\kappa_y^t$, while fixed point iteration diverges in 3D. On the other hand, DDIM inversion yields a reasonable approximation of $\kappa_y^t$ and significantly improves 3D generation quality. We provide visual comparison of the obtained noisy images for each baseline in appendix D.

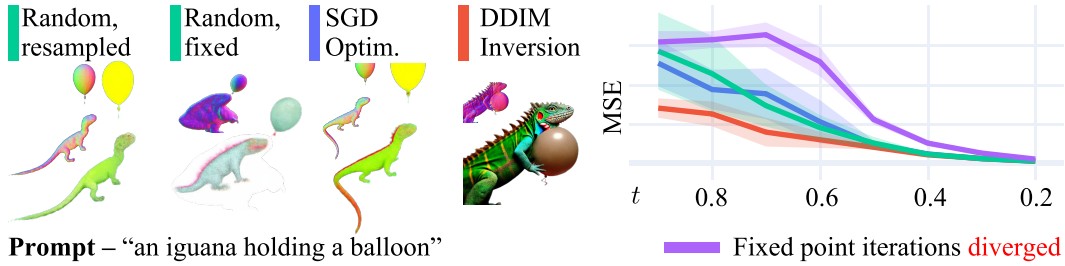

**Prompt** – "an iguana holding a balloon"

Figure 9: The ablation study of different $\kappa_y^t$ choices in our algorithm. We show the obtained 3D generations on the left (Fixed Point Iteration diverges), and the numerical error in eq. (8) induced for each timestep on the right. The resampled and fixed noise strategies produce the same error.

**CFG for inversion.** [33, 34] report that DDIM inversion accumulates big numerical error for CFG $\gamma > 0$. Surprisingly, we find that DDIM inversion for CFG $\gamma_{\text{fwd}} > 0$ can be adequately estimated by running the inversion with negative CFG $\gamma_{\text{inv}} = -\gamma_{\text{fwd}}$. Figure 10 compares 3D inversion strategies qualitatively and quantitatively. Naïvely taking $\gamma_{\text{inv}} = \gamma_{\text{fwd}} = 7.5$ yields the biggest numerical error, while other strategies perform on par. 3D generations, illustrated on the right, show that $\gamma_{\text{inv}} = \gamma_{\text{fwd}} = 7.5$ introduces excessive numerical errors, causing generation to drift in a random direction. The best parameters (as we demonstrate in appendix E) for 2D inversion ($\gamma_{\text{inv}} = \gamma_{\text{fwd}} = 0$) fail to converge in 3D as there is not enough guidance toward a class sample. Introducing guidance only on the forward pass with $\gamma_{\text{inv}} = 0, \gamma_{\text{fwd}} = 7.5$ solves the problem, and the algorithm generates the desired 3D shape, but constantly adding CFG on each step over-saturates the image. Note in that configuration the inversion is performed unconditionally from the text prompt, matching ISM [29]. As we can see, prompt conditioning is an important component in eq. (8), and using $\gamma_{\text{inv}} = -7.5, \gamma_{\text{fwd}} = 7.5$ cancels the over-saturation and produces accurate 3D generations.

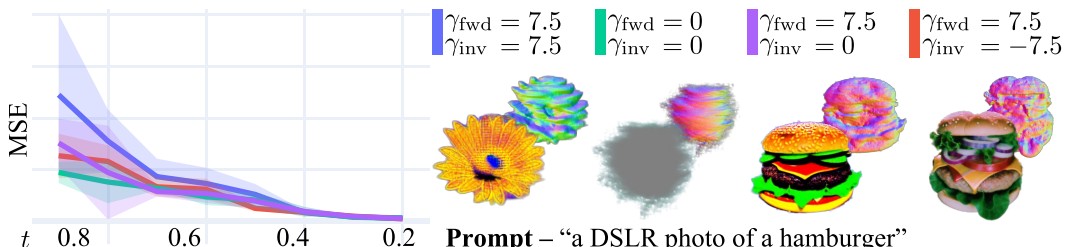

**Prompt** – "a DSLR photo of a hamburger"

Figure 10: Comparing DDIM inversion strategies. Left: Numerical error in eq. (8) from the inferred noise. Right: Generations for different strategies of using CFG values for denoising and inversion.

**Number of steps for DDIM inversion.** Figure 11 ablates the number of steps required for DDIM inversion. We use $n = 10$ as it provides a good balance between generation quality and speed.

**Prompt** – "photograph of a baby raccoon holding a hamburger"

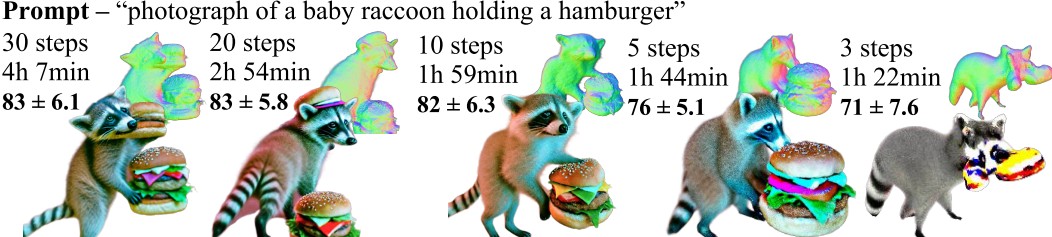

30 steps
4h 7min
**83 ± 6.1**

20 steps
2h 54min
**83 ± 5.8**

10 steps
1h 59min
**82 ± 6.3**

5 steps
1h 44min
**76 ± 5.1**

3 steps
1h 22min
**71 ± 7.6**

Figure 11: Ablation study of the number of inversion steps. For each configuration we report an average run time and CLIP IQA "quality" computed on 43 different prompts.

## 7 Conclusion, Limitations, and Future Work

Helping explain the discrepancy between high-quality image generation with diffusion models and the blurry, over-saturated 3D generation of SDS, our derivation exposes how the strategies are reparameterizations of one another up to a single term. Our proposed algorithm SDI closes the gap between these methods, matching the performance of the two in 2D and significantly improving 3D results. The ablations show that DDIM inversion adequately approximates the correct noise term, and adding it to SDS significantly improves visual quality. The results of SDI match or surpass state-of-the-art 3D generations, all without separate diffusion models or additional generation steps.

Some limitations of our algorithm motivate future work. While we improve the sample quality of each view, 3D consistency between views remains challenging; as a result, despite the convexity loss, our algorithm occasionally produces flat or concave "billboards." A possible resolution might involve supervision with pre-trained depth or normal estimators. A related problem involves content drift from one view to another. Since there was no 3D supervision, there is little to no communication between opposite views, which can lead to inconsistent 3D assets. Stronger view conditioning, multi-view supervision, or video generation models might resolve this problem. Finally, score distillation is capped by the performance of the underlying diffusion model and is hence prone to reproduce similar "hallucinations" (e.g., text and limbs anomalies); the algorithm inherits the biases of the 2D diffusion model and can produce skewed distributions. Appendix H.1 demonstrates typical failure cases.

**Potential broader impacts.** Our work extends existing 2D diffusion models to generation in 3D settings. As such, SDI could marginally improve the ability of bad actors to generate deepfakes or to create 3D assets corresponding to real humans to interact with in virtual environments or games; it also inherits any biases present in the 2D model. While this is a critical problem in industry, our work does not explicitly focus on this use case and, in our view, represents a negligible change in such risks, as highly convincing deepfake tools are already widely available.

## Acknowledgments

The authors thank Lingxiao Li and Chenyang Yuan for their thoughtful insights and feedback.

Artem Lukoianov acknowledges the generous support of the Toyota–CSAIL Joint Research Center.

Vincent Sitzmann was supported by the National Science Foundation under Grant No. 2211259, by the Singapore DSTA under DST00OECI20300823 (New Representations for Vision and 3D Self-Supervised Learning for Label-Efficient Vision), by the Intelligence Advanced Research Projects Activity (IARPA) via Department of Interior/Interior Business Center (DOI/IBC) under 140D0423C0075, by the Amazon Science Hub, and by IBM.

The MIT Geometric Data Processing Group acknowledges the generous support of Army Research Office grants W911NF2010168 and W911NF2110293, of National Science Foundation grant IIS-2335492, from the CSAIL Future of Data program, from the MIT–IBM Watson AI Laboratory, from the Wistron Corporation, and from the Toyota–CSAIL Joint Research Center.

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

# Appendix

## A Implementation details.

In this section, we provide implementation details for our algorithm. Figure 12 provides a side-by-side comparison of SDS and our algorithm.

---

| **Algorithm 1** Dreamfusion (SDS) | **Algorithm 2** Ours (SDI) |
|---|---|
| **Input:** $\psi \in \mathbb{R}^N$ - parametrized 3D shape | **Input:** $\psi \in \mathbb{R}^N$ - parametrized 3D shape |
| $\quad \mathcal{C}$ - set of cameras around the 3D shape | $\quad \mathcal{C}$ - set of cameras around the 3D shape |
| $\quad y$ - text prompt | $\quad y$ - text prompt |
| $\quad g : \mathbb{R}^N \times \mathcal{C} \to \mathbb{R}^{n \times n}$ - differentiable renderer | $\quad g : \mathbb{R}^N \times \mathcal{C} \to \mathbb{R}^{n \times n}$ - differentiable renderer |
| $\quad \epsilon_\theta^{(t)} : \mathbb{R}^{n \times n} \to \mathbb{R}^{n \times n}$ - trained diffusion model | $\quad \epsilon_\theta^{(t)} : \mathbb{R}^{n \times n} \to \mathbb{R}^{n \times n}$ - trained diffusion model |
| **Output:** 3D shape $\psi$ of $y$ | **Output:** 3D shape $\psi$ of $y$ |
| $\quad$ **procedure** DREAMFUSION($y$) | $\quad$ **procedure** OURS($y$) |
| $\quad\quad$ **for** $i$ in range($n\_iters$) **do** | $\quad\quad$ **for** $i$ in range($n\_iters$) **do** |
| $\quad\quad\quad$ $t \leftarrow$ Uniform$(0, 1)$ | $\quad\quad\quad$ $t \leftarrow 1 - i/n\_iters$ |
| $\quad\quad\quad$ $c \leftarrow$ Uniform$(\mathcal{C})$ | $\quad\quad\quad$ $c \leftarrow$ Uniform$(\mathcal{C})$ |
| $\quad\quad\quad$ $\epsilon \leftarrow$ Normal$(0, I)$ | $\quad\quad\quad$ $\epsilon \leftarrow \kappa_y^{t+\tau}(g(\psi, c))$ |
| $\quad\quad\quad$ $x_t \leftarrow \sqrt{\alpha(t)} g(\psi, c) + \sqrt{1 - \alpha(t)} \epsilon$ | $\quad\quad\quad$ $x_t \leftarrow \sqrt{\alpha(t)} g(\psi, c) + \sqrt{1 - \alpha(t)} \epsilon$ |
| $\quad\quad\quad$ $\nabla_\psi \mathcal{L}_{SDS} = \sigma(t) \left[ \epsilon_\theta^{(t)}(x_t, y) - \epsilon \right] \frac{\partial g}{\partial \psi}$ | $\quad\quad\quad$ $\nabla_\psi \mathcal{L}_{SDS} = \sigma(t) \left[ \epsilon_\theta^{(t)}(x_t, y) - \epsilon \right] \frac{\partial g}{\partial \psi}$ |
| $\quad\quad\quad$ Backpropagate $\nabla_\psi \mathcal{L}_{SDS}$ | $\quad\quad\quad$ Backpropagate $\nabla_\psi \mathcal{L}_{SDS}$ |
| $\quad\quad\quad$ SGD update on $\psi$ | $\quad\quad\quad$ SGD update on $\psi$ |

Figure 12: Comparison of the original SDS algorithm and our proposed changes.

Our implementation uses the following choice of $\kappa_y^t$:

$$H(t) = 0.3\sqrt{1 - \alpha(t)}\epsilon_H \text{ , where } \epsilon_H \sim \mathcal{N}(0, I)$$
$$\kappa_y^t(x) = \text{ddim\_inversion}(x, t) + H(t) \tag{11}$$

### A.1 Timesteps.

In eq. (10), the added noise inverts eq. (8) up to noise level $t$, but the denoising step happens to slightly lower time $t - \tau$ ($t_1$ and $t_2$ in fig. 4, right). Intuitively, $\tau$ controls the effective step size of the denoising (or image improvement) process. To accommodate this, we maintain a global time variable $t$ that linearly decays for all the views. Then, on each update step we run DDIM inversion up to $t + \tau$, where $\tau$ is a small constant. In practice, we did not find the algorithm to be sensitive to this constant and sample $\tau \sim U(0, \frac{1}{30})$, where $\frac{1}{30}$ is a typical step size in DDIM.

### A.2 Geometry regularization.

The *Janus problem* is a common issue reported across multiple score distillation works [5, 11, 12], wherein the model produces frontal views all around the shape due to a bias for views dominant in the training data. SDS [5] tackles this problem by augmenting the prompts with information about the view direction. Our method's variance reduction, however, enabled the use of lower CFG values $\gamma = 7.5$ (instead of 100 for SDS), and as shown in fig. 3, with smaller CFG values, the diffusion model tends to ignore certain parts of the prompts (like "colored" for $\gamma < 10$ in fig. 3). Thus in our algorithm we observe the diffusion model to be less attentive to the prompt augmentation yielding a stronger Janus problem compared to SDS; we also see the same behavior for other baselines that reduce CFG. To address this issue, we use Perp-Neg [36] and add a small entropy term $\sim \mathcal{N}(0, 0.3\sqrt{1 - \alpha(t)}I)$ (eq. (11)) to the inverted noise to reduce mode-seeking behavior as in [44].

Another common issue in Score Distillation is the *hollow face illusion* [46], which is a tendency to create holes in the geometry of a visually convex shape. To address this problem, we use a normal map extracted from the volume as described in [5] and downscaled to $8 \times 8$ resolution so that only low frequencies are penalized. Then, we calculate the sine between adjacent normals (from left to

right and top to bottom) and penalize it to be negative. We activate this loss for the first $40\%$ of the training steps with weight $\alpha_{\text{convexity}} = 0.1$ to break the symmetry.

Appendix H.1 demonstrates examples of typical failure cases, including the Janus problem and the hollow face illusion.

### A.3 System details.

We implement our algorithm in `threestudio` [43] on top of SDS [5]. We use Stable Diffusion 2.1 [1] as the diffusion model. For volumetric representation we use InstantNGP [8]. Instead of randomly sampling time $t$ as in SDS, we maintain a global parameter $t$ that linearly decays from 1 to 0.2 (lower time values do not have a significant contribution). Next, for each step we render a $512 \times 512$ random view and infer $\kappa^{t+\tau}$ by running DDIM inversion for $\text{int}(10t)$ steps—i.e. we use 10 steps of DDIM inversion for $t = 1$ and linearly decrease it for smaller $t$. We use NVIDIA A6000 GPUs and run each generation for $10k$ steps with learning rate of $10^{-2}$, which takes approximately 2 wall clock hours per shape generation.

### A.4 Prompts used in the quantitative evaluation

```
"A car made out of sushi"
"A delicious croissant"
"A small saguaro cactus planted in a clay pot"
"A plate piled high with chocolate chip cookies"
"A 3D model of an adorable cottage with a thatched roof"
"A marble bust of a mouse"
"A ripe strawberry"
"A rabbit, animated movie character, high detail 3d mode"
"A stack of pancakes covered in maple syrup"
"An ice cream sundae"
"A baby bunny sitting on top of a stack of pancakes"
"Baby dragon hatching out of a stone egg"
"An iguana holding a balloon"
"A blue tulip"
"A cauldron full of gold coins"
"Bagel filled with cream cheese and lox"
"A plush dragon toy"
"A ceramic lion"
"Tower Bridge made out of gingerbread and candy"
"A pomeranian dog"
"A DSLR photo of Cthulhu"
"Pumpkin head zombie, skinny, highly detailed, photorealistic"
"A shell"
"An astronaut is riding a horse"
"Robotic bee, high detail"
"A sea turtle"
"A tarantula, highly detailed"
"A DSLR photograph of a hamburger"
"A DSLR photo of a soccer ball"
"A DSLR photo of a white fluffy cat"
"A DSLR photo of a an old man"
"Renaissance-style oil painting of a queen"
"DSLR photograph of a baby racoon holding a hamburger, 80mm"
"Photograph of a black leather backpack"
"A DSLR photo of a freshly baked round loaf of sourdough bread"
"A DSLR photo of a decorated cupcake with sparkling sugar on top"
"A DSLR photo of a dew-covered peach sitting in soft morning light"
"A photograph of a policeman"
"A photograph of a ninja"
"A photograph of a knight"
"An astronaut"
```

```
"A photograph of a firefighter"
"A Viking panda with an axe"
```

# B  Comparison with Interval Score Matching

In this section, we extensively compare our algorithm with Interval Score Matching (ISM) [29].

**Theoretical assumptions.**   ISM empirically observes that "pseudo-GT" images used to guide Score Distillation Sampling (SDS) are sensitive to their input and that the single-step generation of pseudo-GT yields over-smoothing. From these observations, starting with SDS guidance, ISM adds DDIM inversion and multi-step generation to empirically improve the stability of the guidance. In contrast, our work starts with 2D diffusion sampling to re-derive score-distillation guidance and motivate improvements. That is, our work formally connects SDS to well-justified techniques in 2D sampling.

**DDIM inversion.**   In ISM, the empirically motivated DDIM inversion is at the basis of the derivation of the final update rule. We suggest a general form of the noise term eq. (8), for which *DDIM inversion is just one possible solution*. Our theoretical insights are agnostic to particular algorithms of root-finding, which makes it possible to use more efficient solutions in future research (e.g. train diffusion models as invertible maps to sample noise faster).

**Guidance term.**   The update rules provided in our eq. (10) and ISM's eq.17 have two main differences:

1. **Full vs. interval noise.** Assuming DDIM inversion finds a perfect root of our stationary equation, ISM's update rule can take a similar form to ours eq. (10) (interval noise is equal to $\kappa$ if eq. (8) is satisfied). However, as shown in fig. 9, DDIM inversion does not find a perfect root, and thus the two forms are not equivalent. Our theory shows that the full noise term is more accurate. We also show the effect of choosing one term vs. another in fig. 14.
2. **Conditional vs. unconditional inversion.** Our derivation hints that the roots of eq. (8) are prompt-dependent, motivating our use of conditional DDIM inversion (not used in ISM). Our fig. 10 shows how unconditional inversion yields over-saturation.

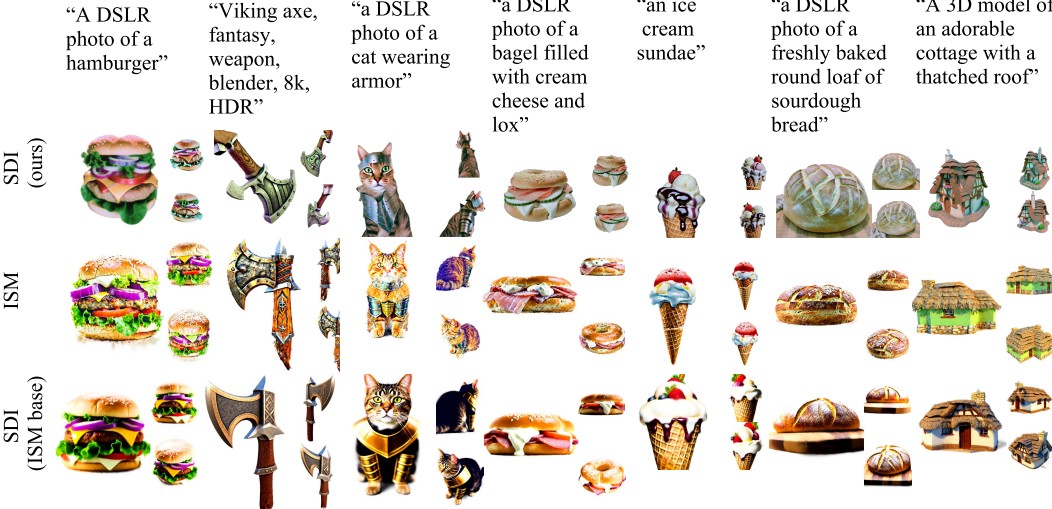

Figure 13:  Comparison of SDI (ours, top) ISM (middle), and SDI in ISM's code base (ours, bottom). To eliminate factors such as 3D representation and initialization, we re-implement our algorithm in the official codebase of ISM. See fig. 14 for the effect of each change made.

**Practical Results.**   To control for different design choices (Gaussian Splattings in ISM vs. NeRF in ours, etc.) we re-implemented our algorithm in the code base of ISM with only the minimal changes discussed in the previous section.

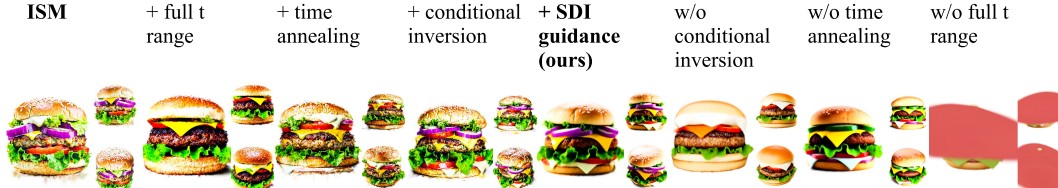

| ISM | + full t range | + time annealing | + conditional inversion | **+ SDI guidance (ours)** | w/o conditional inversion | w/o time annealing | w/o full t range |

Figure 14: Ablation of changes made to the code base of Interval Score Matching (ISM). Prompt "A DSLR photo of a hamburger". We start with the official implementation of ISM and add only four changes to match ours (all the hyper-parameters remain tuned for ISM): 1. instead of sampling t from U(0.02, 0.5) we sample it from U(0.1, 0.98); 2. introduce linear time annealing; 3. use conditional DDIM inversion with negative guidance; 4. use SDI guidance, i.e. second term in the guidance difference is the full noise added to

Figure 13 provides a qualitative comparison using the prompts and settings in ISM's code. Figure 14 shows the effect of each change made to ISM guidance. Table 2 the quantitative comparison (ISM code base for both).

Table 2: Quantitative comparisons between SDI (ours) and ISM.

| Method | CLIP Score (↑) | CLIP IQA (%, ↑) | | | ImageReward (↑) | Time | VRAM |
|---|---|---|---|---|---|---|---|
| | | "quality" | "sharpness" | "real" | | | |
| ISM, 5k steps | **28.60±2.03** | 0.85±0.02 | 0.98±0.01 | 0.98±0.01 | -0.52±0.48 | 45min | 15.4GB |
| SDI (ours), 5k steps | 28.47±1.29 | **0.88±0.03** | **0.99±0.00** | 0.98±0.01 | **-0.30±0.32** | 43min | 15.4GB |

**Summary.** We bridge the gap between experimentally-based score distillation techniques and theoretically-justified 2D sampling. Both SDS and ISM can be seen as different approaches to finding roots of eq. (8). This theoretical insight allows us to modify both ISM and SDS, reducing over-saturation for the first and improving general quality of the second.

## C  Diversity of the generations

Next, we explore the diversity of the results generated by our method. Figure 15 depicts generated shapes for different seeds and prompts. Generations with our method exhibit a certain degree of diversity in local details, but are mostly the same at a coarse level. This is similar to the reported results in other works on score distillation [5, 11]. We plan to address this problem in future research.

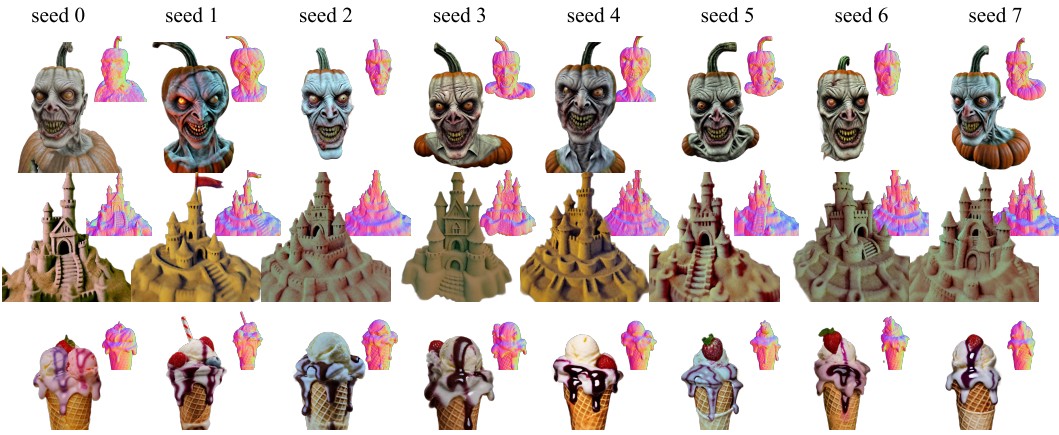

Figure 15: Examples of generations with SDI (ours) for different seed values. Prompts are "Pumpkin head zombie, skinny, highly detailed, photorealistic" (top row), "A highly-detailed sandcastle" (middle row), "An ice cream sundae" (bottom row).

Additionally, we noticed a minimal diversity in human generations. For example, fig. 16 demonstrates generated shapes for 3 prompts related to different professions.

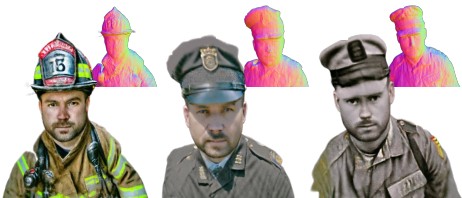

Figure 16: Examples of limited diversity across different human-related prompts: "a photo of a firefighter", "a photo of a policeman", "a photo of a soldier."

## D    Visual comparison of noise patterns

In this section, we provide additional examples of the noise samples obtained with different $\kappa_y^t$. As can be seen from fig. 17, optimization with SGD almost does not change the noise pattern compared to random sampling. Fixed-point iterations drift in a notable but counterproductive direction, and the single-step prediction from the obtained noisy images (depicted on the right side) degrades. DDIM inversion with $\gamma_{\text{inv}} = 7.5$ and $\gamma_{\text{inv}} = -7.5$ produces similar noise patterns located around the "meaningful" parts of the image; however, $\gamma_{\text{inv}} = -7.5$ results in noise patterns that are much more compatible with the single-step denoising procedure used in SDS, leading to much more accurate details appearing for earlier $t$.

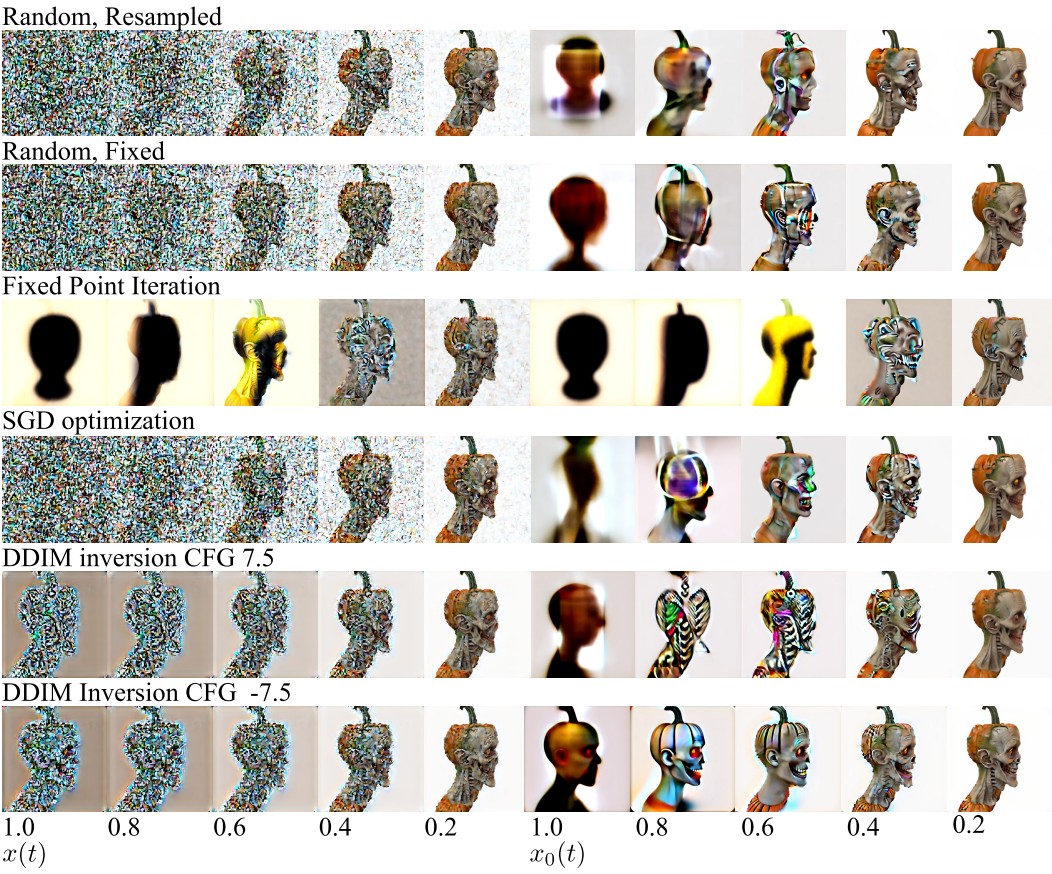

Figure 17: Examples of noisy images (left part) obtained for different choices of $\kappa_y^t$. On the right part, a single step denoised images are presented for the corresponding noisy images.

## E CFG for inversion in 2D

In this section, we provide additional experiments testing the DDIM inversion strategy and study the performance of different combinations of the CFG value on the inversion stage ($\gamma_{\text{inv}}$) and on the forward pass ($\gamma_{\text{fwd}}$). In particular, in fig. 18 we at first generate an image with 30 steps of DDIM (left-most image) and $\gamma = 7.5$. Then we run DDIM in the opposite direction of $t$ with $\gamma_{\text{inv}}$ all the way to $t = 1$. Finally, we use the obtained noise sample as an initialization for a new DDIM inference with $\gamma_{\text{fwd}}$. Ideally, the final image will correspond to the input of the inversion algorithm.

Figure 18 shows that when inverting the DDIM trajectory starting with a clean image, the best inversion strategy uses $\gamma_{\text{inv}} = 0$ and then regenerates the image via DDIM with $\gamma_{\text{fwd}} = 0$. [34] suggests this approach, which inverts the computation almost perfectly. Other approaches introduce bias and only approximate the image on the forward pass.

As we have seen in section 6.2, in 3D, the story is different: $\gamma_{\text{inv}} = \gamma_{\text{fwd}} = 0$ fails to generate any meaningful shape. We explain this behavior by the fact that in our 2D experiment, the original image contains class information; inverting with $\gamma_{\text{inv}} = 0$ encodes this information into the noise and thus does not need additional CFG during reconstruction. In 3D generation, however, the initial NeRF does not contain class information and needs bigger CFG on the forward pass. Additionally, fig. 18 demonstrates the inversion quality of an entire DDIM trajectory, whereas eq. (10) we are interested in inverting a single-step denoising only. what might explain the fact that seemingly bad inversion quality of $\gamma_{\text{fwd}} = -\gamma_{\text{inv}} = 7.5$ yields much better results in 3D.

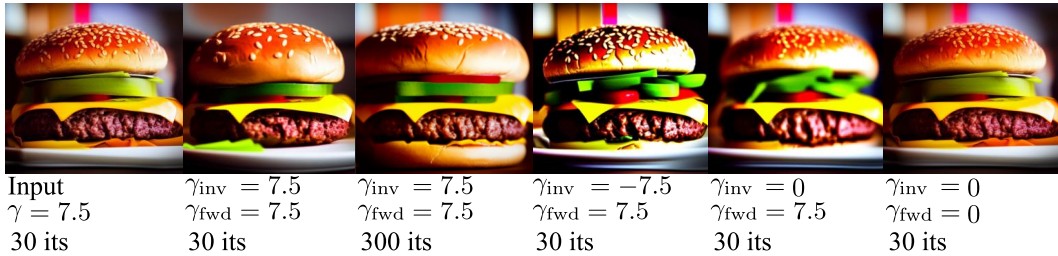

| Input | $\gamma_{\text{inv}} = 7.5$ | $\gamma_{\text{inv}} = 7.5$ | $\gamma_{\text{inv}} = -7.5$ | $\gamma_{\text{inv}} = 0$ | $\gamma_{\text{inv}} = 0$ |
|---|---|---|---|---|---|
| $\gamma = 7.5$ | $\gamma_{\text{fwd}} = 7.5$ | $\gamma_{\text{fwd}} = 7.5$ | $\gamma_{\text{fwd}} = 7.5$ | $\gamma_{\text{fwd}} = 7.5$ | $\gamma_{\text{fwd}} = 0$ |
| 30 its | 30 its | 300 its | 30 its | 30 its | 30 its |

Figure 18: Comparison of different DDIM inversion strategies in 2D. Here, we compare the quality of inversion, i.e., the forward pass is the entire DDIM trajectory.

## F ODE derivation

Our derivation in section 4 manipulates the time-stepping procedure of DDIM, but a similar argument applies to the ODE version of the method. In particular, one can obtain an alternative form of eq. (10) by reparametrizing the ODE eq. (1). To do the change of variables in eq. (1) to eq. (5) we need to differentiate it with respect to $t$ and evaluate $\frac{d\bar{x}(t)}{dt}$. Direct differentiation gives us:

$$\frac{dx_0(t)}{dt} = \frac{d\bar{x}(t)}{dt} - \epsilon_\theta^{(t)}\left(\frac{\bar{x}(t)}{\sqrt{\sigma(t)^2+1}}, y\right)\frac{d\sigma(t)}{dt} - \sigma(t)\frac{d}{dt}\epsilon_\theta^{(t)}\left(\frac{\bar{x}(t)}{\sqrt{\sigma(t)^2+1}}, y\right). \tag{12}$$

Solving for $\frac{d\bar{x}(t)}{dt}$ and merging with eq. (1), we get:

$$\epsilon_\theta^{(t)}\left(\frac{\bar{x}(t)}{\sqrt{\sigma(t)^2+1}}, y\right)\frac{d\sigma(t)}{dt} = \frac{dx_0(t)}{dt} + \epsilon_\theta^{(t)}\left(\frac{\bar{x}(t)}{\sqrt{\sigma(t)^2+1}}, y\right)\frac{d\sigma(t)}{dt} + \sigma(t)\frac{d}{dt}\epsilon_\theta^t\left(\frac{\bar{x}(t)}{\sqrt{\sigma(t)^2+1}}, y\right)$$

$$\frac{dx_0(t)}{dt} = -\sigma(t)\frac{d}{dt}\epsilon_\theta^{(t)}\left(\frac{\bar{x}(t)}{\sqrt{\sigma(t)^2+1}}, y\right)$$

$$= -\sigma(t)\frac{d}{dt}\epsilon_\theta^{(t)}\left(\frac{x_0(t) + \sigma(t)\kappa^t(x_0(t))}{\sqrt{\sigma(t)^2+1}}, y\right)$$

$$= -\frac{d\kappa^t}{dt}(x_0(t)). \tag{13}$$

Or, expressed in other scaling parameters:

$$\frac{dx_0(t)}{dt} = -\sigma(t)\frac{d}{dt}\epsilon_\theta^{(t)}\left(x_0(t) + \sigma(t)\kappa^t(x_0(t)), y\right).$$ (14)

This expression gives us the DDIM's ODE re-parametrized for the dynamics of $x_0(t)$. This ODE is a velocity field that projects a starting image to a conditional distribution learned by the diffusion model. Discretizing this equation leads to eq. (10).

## G  Additional Intuition: Out-of-Distribution Correction

Here we provide an intuition why it is so important to regress the noise $\kappa$ instead of randomly sampling it. In particular, we can look into the data that was used to train $\epsilon_\theta^{(t)}$. For each $t$, the model has seen only clean images with the corresponding amount of noise $\sigma(t)$.

Consider a noisy or blurry image $\hat{x}_0$. Our goal is to use our updated ODE to improve it and make it look more realistic. For $t$ close to 1, the amount of noise is so big, that for any image $\hat{x}_0$ its noisy version $\hat{x}_0 + \sigma(t)\epsilon$ looks indistinguishable from the training data of the denoiser. This explains why we can successfully use diffusion models to correct out-of-distribution images.

For small $t$, however, the noise levels might be not enough to hide the artifacts of $\hat{x}_0$, so the image becomes out-of-distribution, and the predictions of the model become unreliable. We hypothesize, that if instead of randomly sampling $\epsilon$ like in Dreamfusion, we can find such a noise sample, that under the same noise level $t$ the artifacts of $\hat{x}_0$ will be hidden and it will be in-distribution of the denoiser. This effectively increases the ranges of $t$ where the predictions of the model are reliable and thus improves the generation quality. This intuition is illustrated in fig. 19.

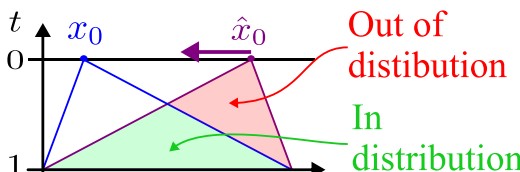

Figure 19: Schematic for out-of-distribution interpretation.

## H  Additional Results

Lastly, we discuss failure cases and showcase additional generations using our methods, along with more baseline comparisons.

### H.1  Failure Cases

We provide illustrations of typical failure cases of our algorithm in Figures 20 to 23.

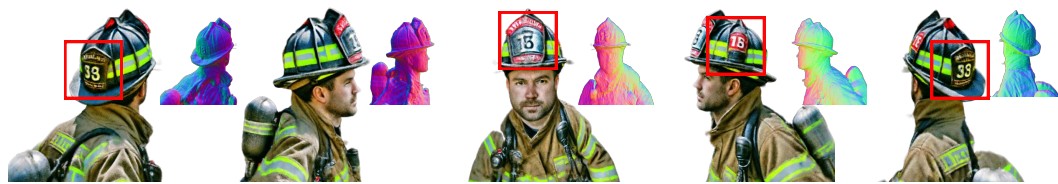

"a photograph of a firefighter"

Figure 20: **Content Drift**: due to a limited communication between the frames sometimes we can observe that certain properties "drift" around the shape. In this particular example the number on the fireman's hat is different from all the views.

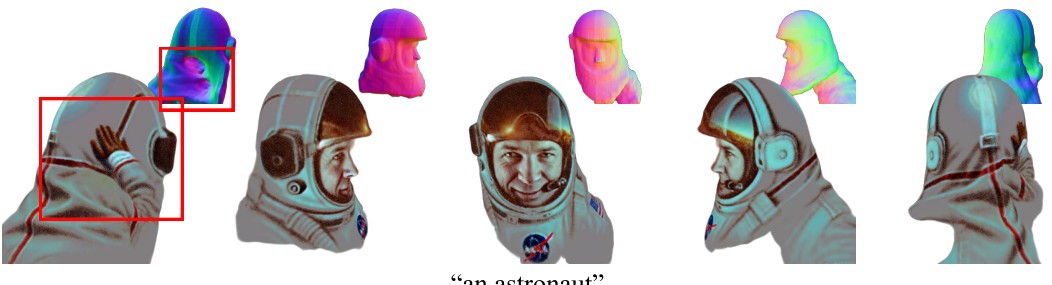

"an astronaut"

Figure 21: **Hollow Face Illusion:** some shapes might be generated with a concave hole in the geometry, while the rendering appears to be convex.

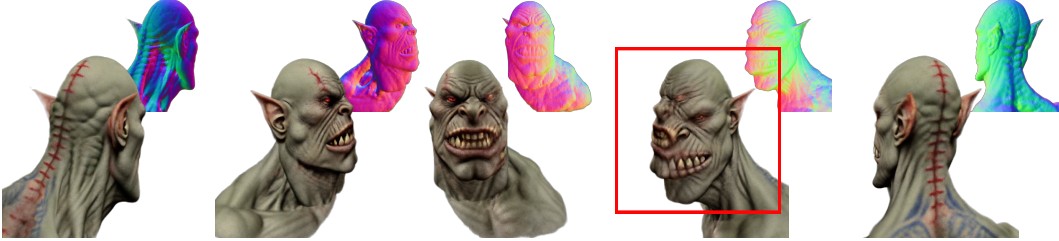

"A portrait photograph of an orc"

Figure 22: **Janus Problem:** note how the orc has two faces that merge on the second to the right view.

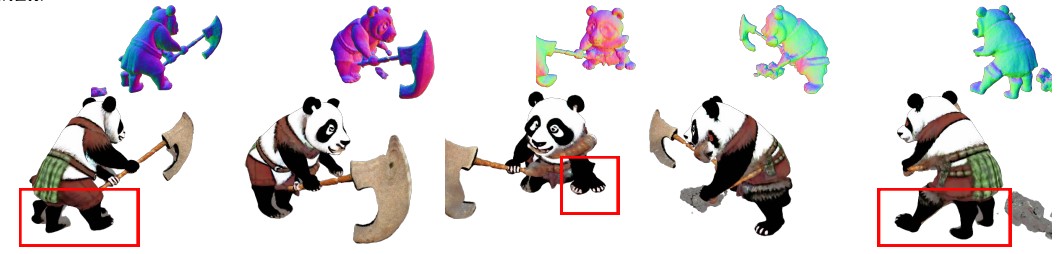

"a Viking panda with an axe"

Figure 23: **Diffusion Anomalies:** performance of our algorithm is limited by the quality of the underlying diffusion model. In this case the generated shape contains a typical failure case of multiple limbs.

## H.2 Additional generations

Figures 24 to 27 provide additional generations produced using our method.

## H.3 Additional comparison with baselines

We provide additional qualitative comparisons with other score distillation algorithms in figs. 28 to 30. We compare with Dreamfusion [5], Magic3D [14], Fantasia3D [47], Stable Score Distillation (StableSD [27], Classifier Score Distillation (ClassifierSD [16]), Noise-Free Score Distillation (NFSD [12]), ProlificDreamer or Variational Score Distillation (VSD [11]), Interval Score Matching (ISM) [29], and HiFA [13].

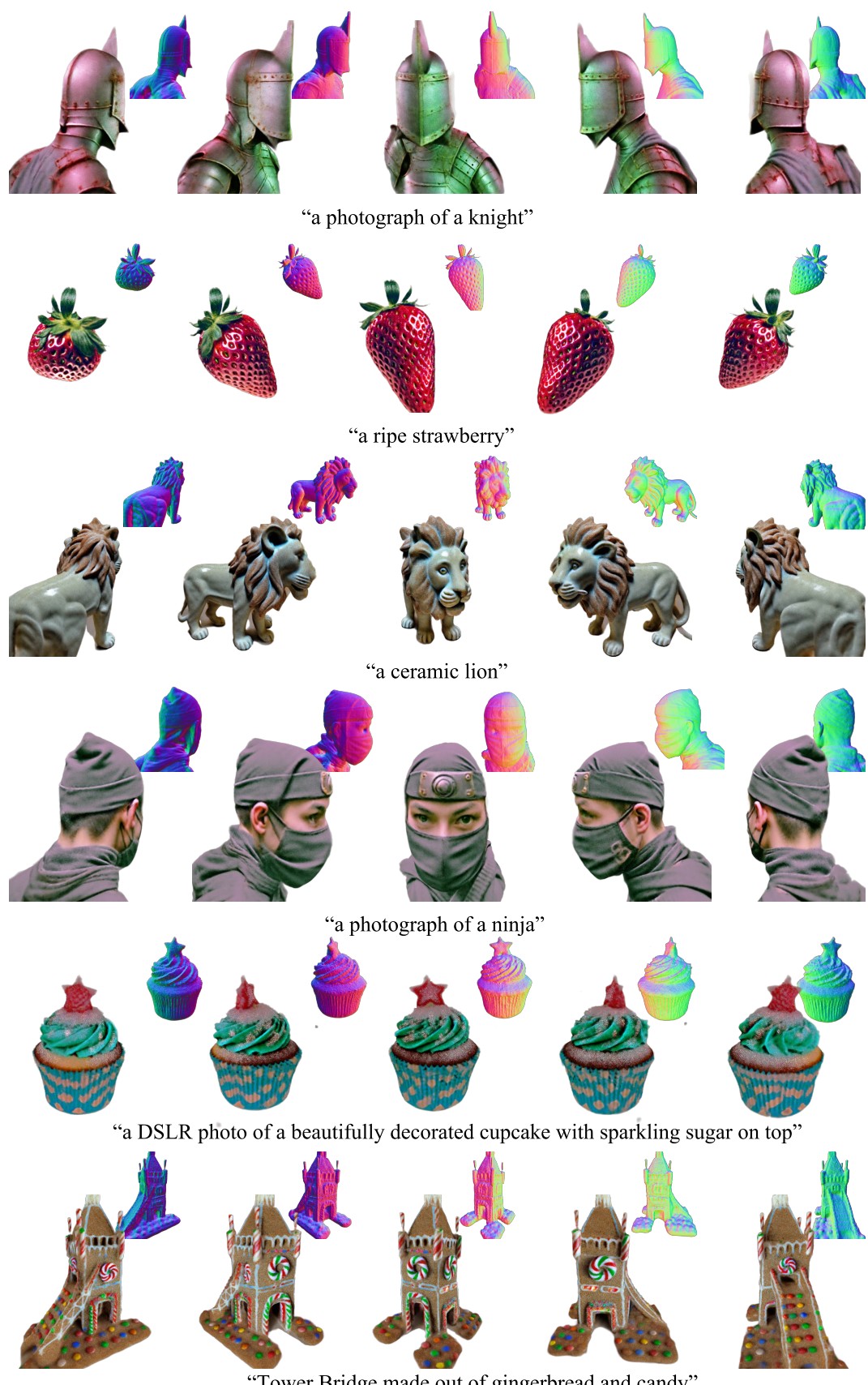

"a photograph of a knight"

"a ripe strawberry"

"a ceramic lion"

"a photograph of a ninja"

"a DSLR photo of a beautifully decorated cupcake with sparkling sugar on top"

"Tower Bridge made out of gingerbread and candy"

Figure 24: Additional generations from our method.

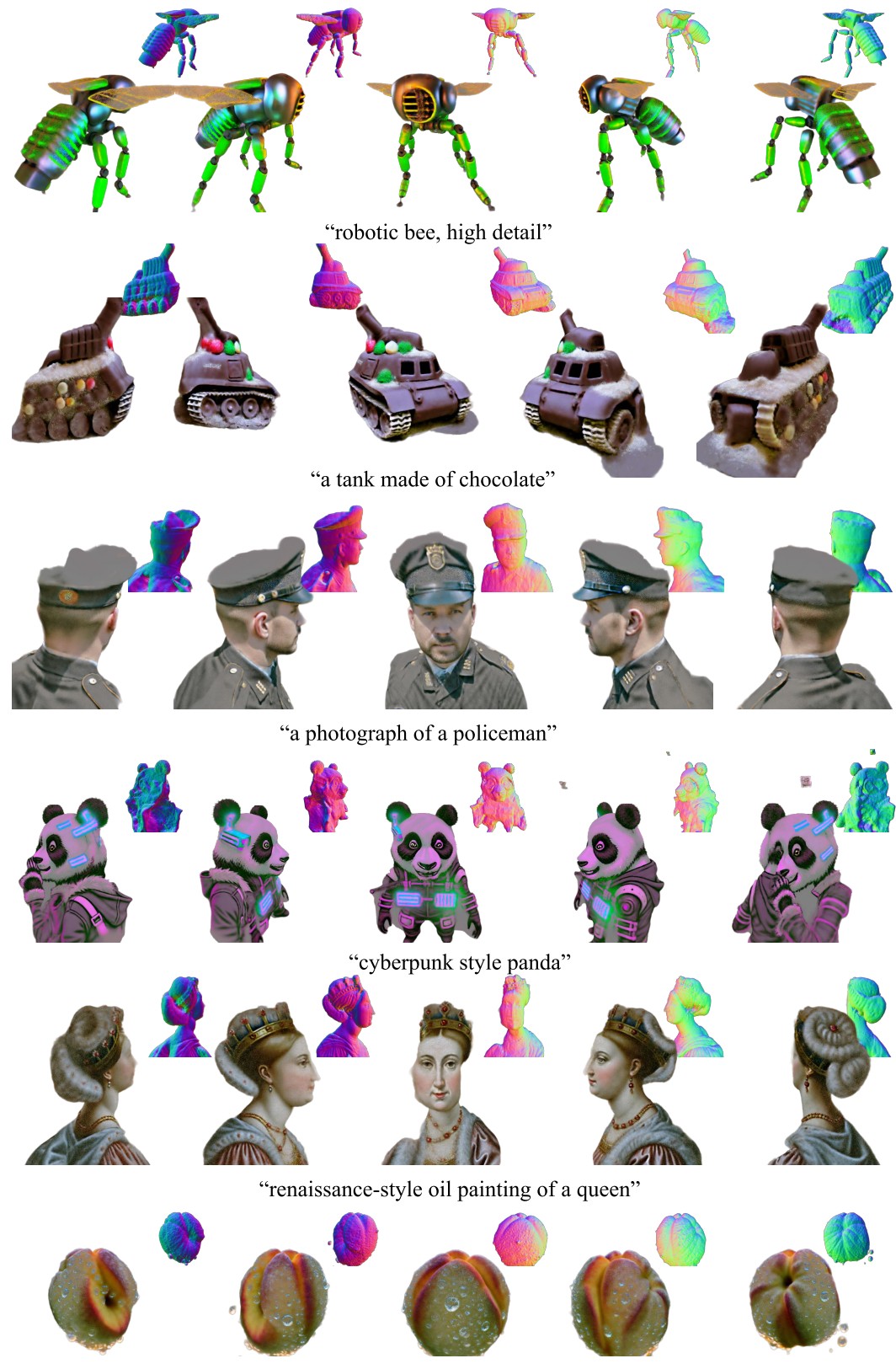

"robotic bee, high detail"

"a tank made of chocolate"

"a photograph of a policeman"

"cyberpunk style panda"

"renaissance-style oil painting of a queen"

"a DSLR photo of a dew-covered peach sitting in soft morning light"

Figure 25: Additional generations from our method.

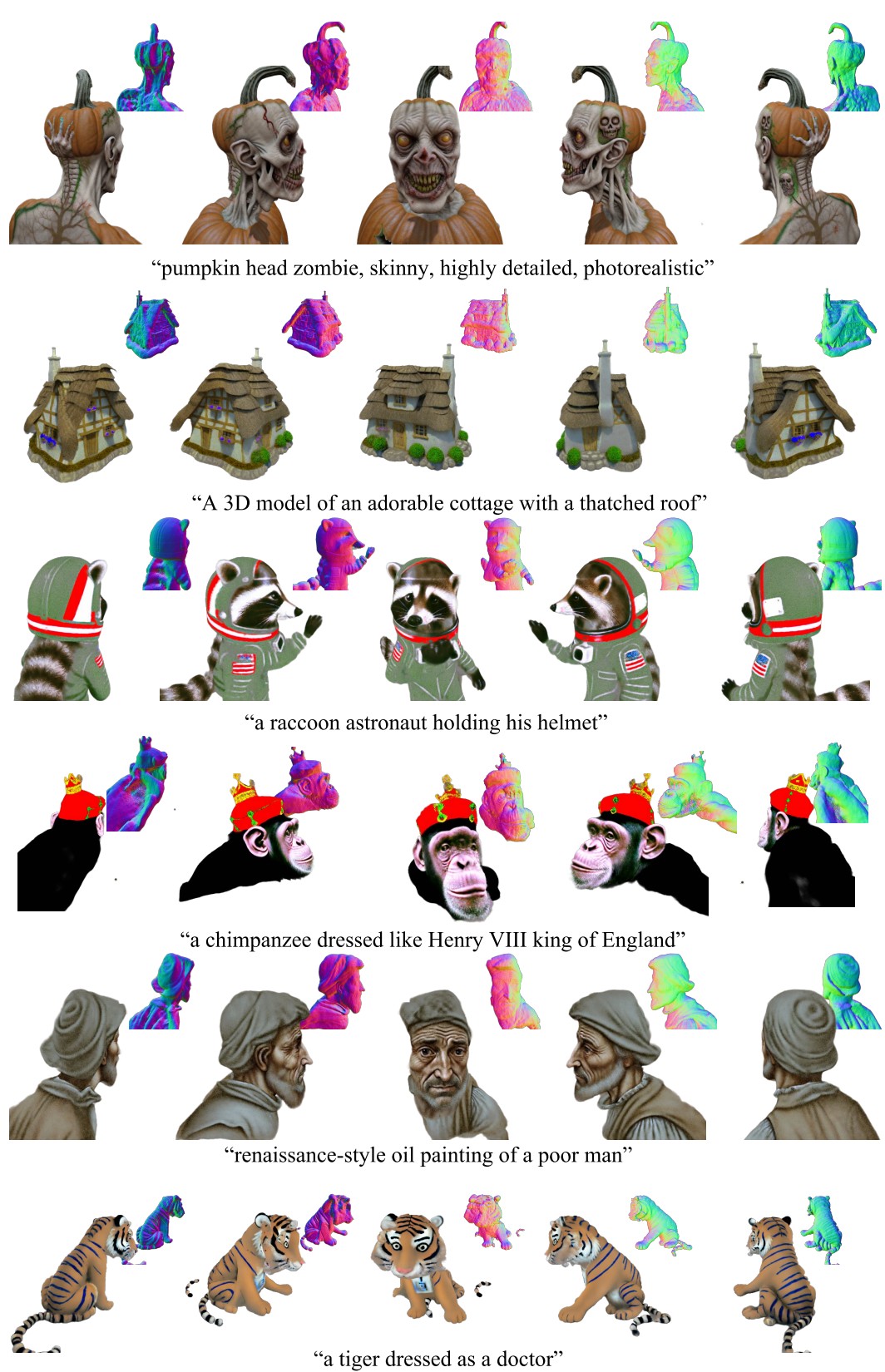

"pumpkin head zombie, skinny, highly detailed, photorealistic"

"A 3D model of an adorable cottage with a thatched roof"

"a raccoon astronaut holding his helmet"

"a chimpanzee dressed like Henry VIII king of England"

"renaissance-style oil painting of a poor man"

"a tiger dressed as a doctor"

Figure 26: Additional generations from our method.

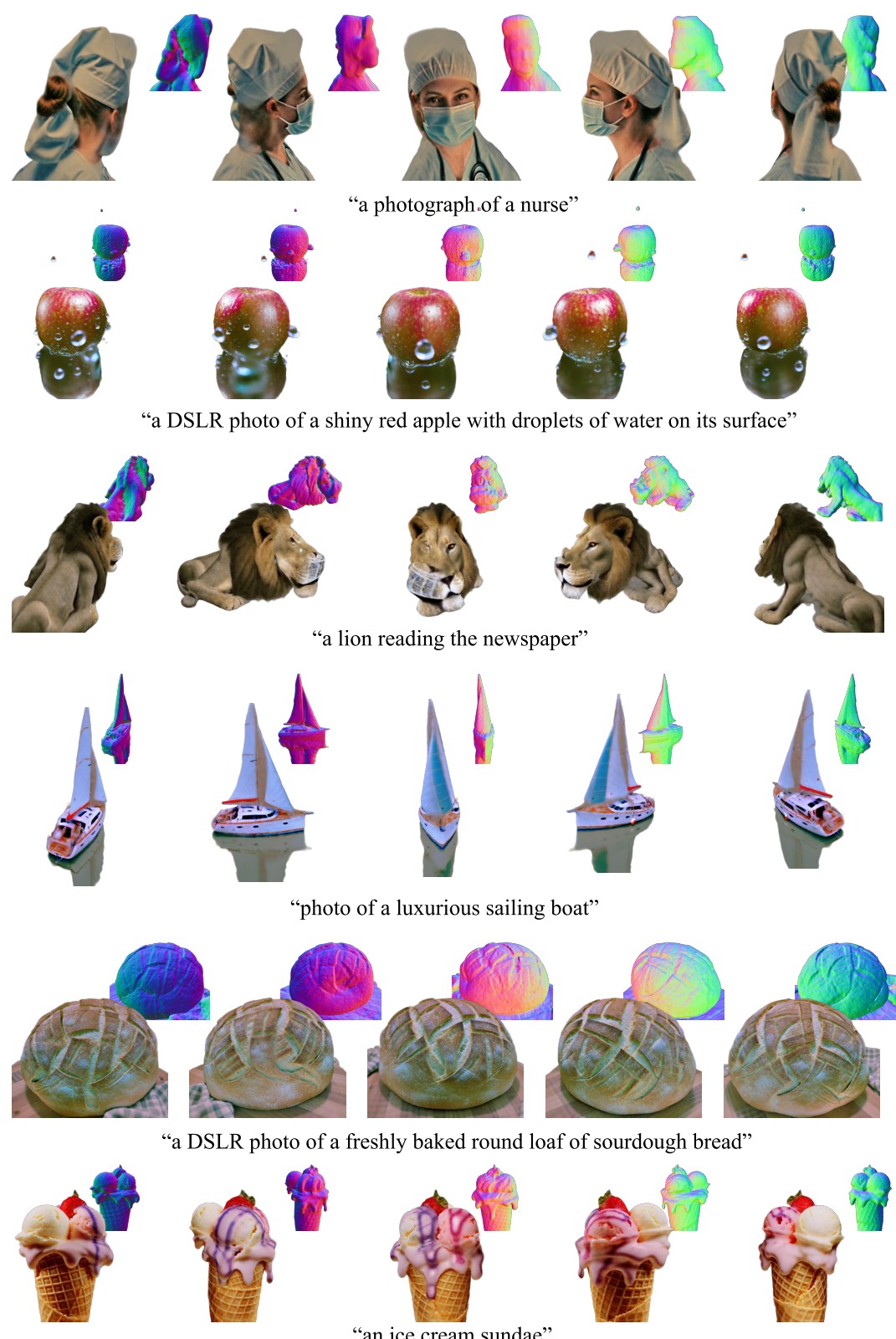

"a photograph of a nurse"

"a DSLR photo of a shiny red apple with droplets of water on its surface"

"a lion reading the newspaper"

"photo of a luxurious sailing boat"

"a DSLR photo of a freshly baked round loaf of sourdough bread"

"an ice cream sundae"

Figure 27: Additional generations from our method.

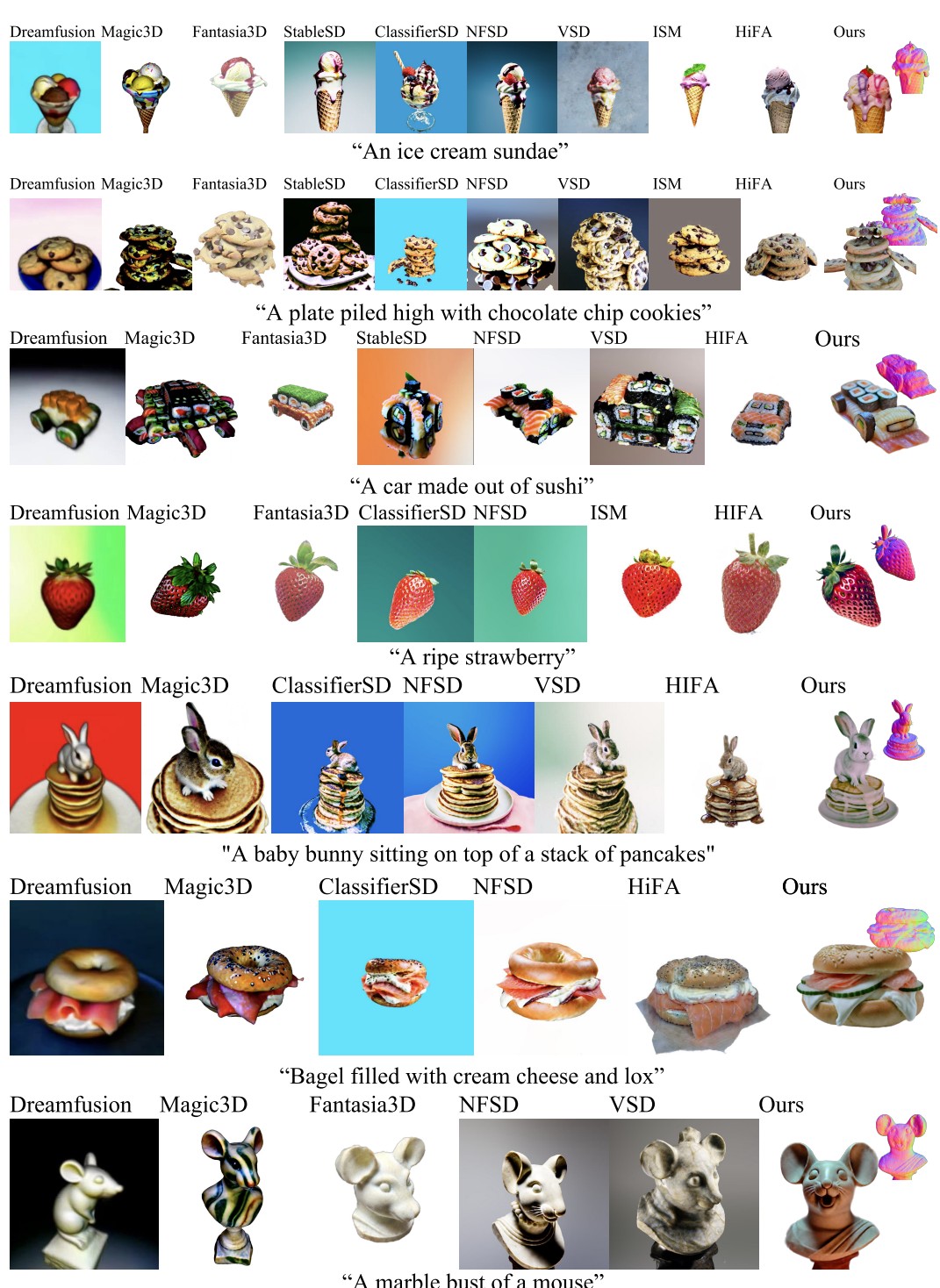

Figure 28: Additional comparisons.

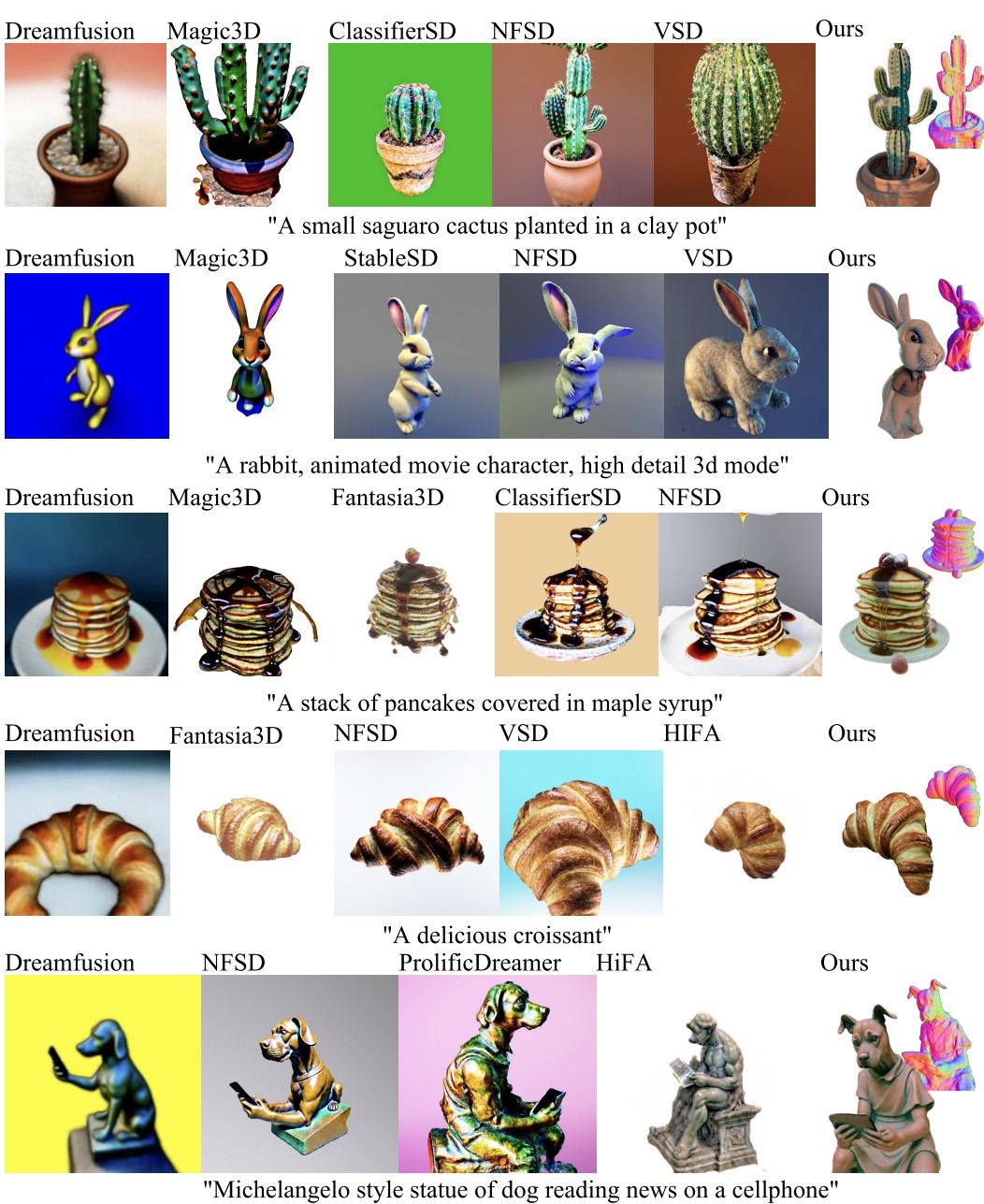

Figure 29: Additional comparisons.

Dreamfusion          Magic3D          NFSD          Ours

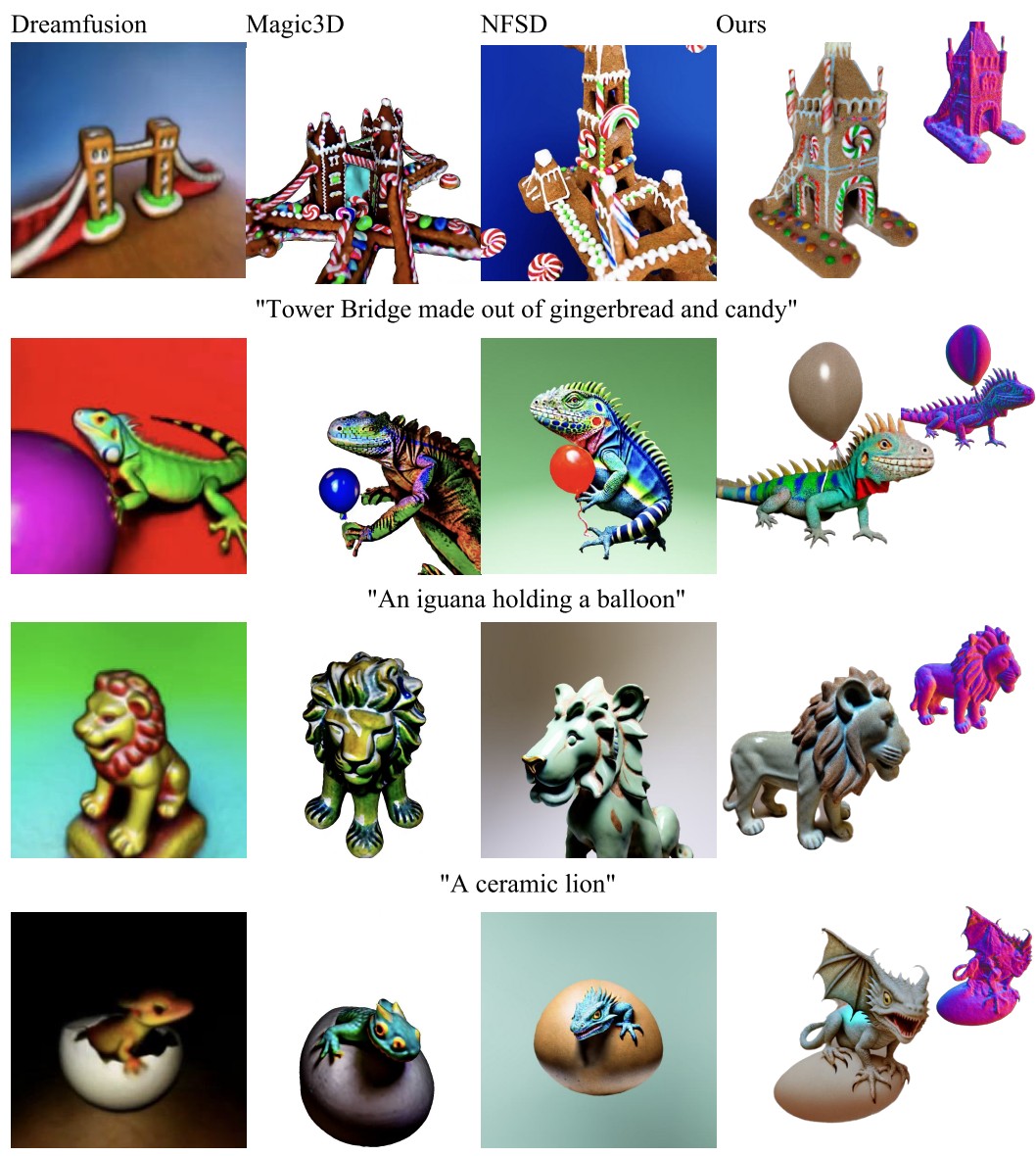

"Tower Bridge made out of gingerbread and candy"

"An iguana holding a balloon"

"A ceramic lion"

"Baby dragon hatching out of a stone egg"

Figure 30: Additional comparisons.

