# OpenReview forum: "Score Distillation via Reparametrized DDIM"
_NeurIPS.cc/2024/Conference — NeurIPS 2024 poster_

### Official Review · Reviewer_ZvgL · 2024-06-29

**Soundness:** 3
**Presentation:** 3
**Contribution:** 3
**Rating:** 7
**Confidence:** 4

**Summary:**

This paper introduces a novel algorithm called Score Distillation via Inversion (SDI) that enhances the quality of 3D shape generation. By inverting Denoising Diffusion Implicit Models (DDIM) at each step and incorporating the initial noise into the estimated score, SDI addresses the over-smoothing and detail loss issues associated with traditional Score Distillation Sampling (SDS) methods in 3D generation. Experimental results demonstrate that SDI significantly improves the quality of 3D shapes, closely matching the quality of 2D image generation, without the need for additional neural network training or multi-view supervision. The work provides valuable insights into understanding and improving 3D asset generation with diffusion models.

**Strengths:**

1. The paper provides a theoretical analysis that links Score Distillation Sampling(SDS) with Denoising Diffusion Implicit Models(DDIM), offering a deeper understanding of the discrepancies between 2D and 3D generation process, which is different from the experimental enhancement made by most of the previous methods.
2. The modification to the existing SDS method is straightforward and does not require training additional networks or multi-view supervision. Furthermore, SDI significantly improves the quality of 3D shape generation, which makes it a practical solution for enhancing 3D generation.
3. This paper provides thorough analysis and present a set of convincing ablation experiments.

**Weaknesses:**

1.	According to the authors’ theory and hypothesis, the better the noise term is Eq. (8) is estimated, the better the 3D generation quality is improved. However, in the ablations about the choice of k(x), SGD optimization doesn’t show significant improvement. Also, this phenomenon can be observed from Figure.10, in which green curve and purple curve present lowest and comparable MSE with orange curve, while the orange curve get the best result. I would appreciate it if the authors can provide some explanations.
2.    One of the paper's key perspective is that SDS introduces high variance. I have read another paper[1] dedicated to  minimizing the variance introduced in SDS, and maybe it's also worthwhile to compare with that.



[1] Tang, B., Wang, J., Wu, Z., & Zhang, L. (2023). Stable score distillation for high-quality 3d generation. ArXiv Preprint ArXiv:2312.09305.

**Questions:**

Please refer to the weaknesses part.

**Limitations:**

Please refer to the weaknesses part.

---

> ### Author Rebuttal · Authors · 2024-08-07
>
> Thank you for the thorough review and recommendations. We provide clarifications below.
> # Numerical error in choices of \kappa
> Thank you for highlighting this effect in the review, we believe a more detailed explanation will indeed benefit the paper. We touch on the intuition around this question in the general response: it is not exactly true that \kappa with a lower error in eq.8 leads to a better 3D generation. It is true, however, that a more precise solution to eq.8 aligns score distillation more closely with DDIM. Other factors like initialization, view consistency, and DDIM’s generative ability also impact 3D quality. Below we address each of the particular questions in detail.
>
> 1. The green configuration in Figure 10 ( $\gamma\_\text{fwd}=1, \gamma_\text{inv}=1$) indeed has the lowest error in eq.8. This is consistent with [1, 2], which claim that DDIM inversion accumulates numerical errors in a conditional generation [1] while achieving almost perfect inversion in an unconditional generation [2]. When we use unconditional generation together with unconditional inversion, we succeed at closely matching the generation guidance to a DDIM trajectory. This trajectory, however, is unconditional. Thus the final 3D shape does not correspond to the desired (or any) prompt: exactly what we see in Figure 10.
>
> 2. As it was correctly noted, the purple and red configurations in Fig.10 (conditional vs. unconditional inversion) exhibit the same approximation error. In practice, both configurations yield detailed 3D shapes (also Fig.10). However, as described in the general reply (see “Guidance Term”), a combination of unconditional inversion and conditional generation leads to the accumulation of saturation in the final shape. Please also refer to Fig.5 of the rebuttal PDF for the visualization of this effect.
>
> 3. Regarding the error of SGD and Random noise in Figure 9, the two lines are mostly within one sigma of each other (please note the translucent areas behind each line signifying standard deviation). The numerical error on the right is computed and averaged across 10 prompts. We did not notice any improvement in 3D generation quality nor in the error induced in eq.8 by performing the stochastic gradient descent on the noise term. The SGD optimization of 10 steps was initialized to a randomly sampled noise image, which might explain the resemblance of the generated shapes to the randomly sampled noise.
> ## Comparison with StableSD
> Thank you for providing a useful reference. We would like to point out that we already have a qualitative comparison with Stable Score Distillation in the appendix (StableSD in section E3).
>
>
> Mathematically, the idea behind this line of work is that the mode-seeking term in score distillation ($\\epsilon\_\theta^t$) exhibits high variance. Thus the second term ($- \epsilon$ in SDS or the LoRA in VSD[3]) reduces the variance of the first. A similar idea was also introduced in SteinDreamer [4], where authors numerically analyze variance in SDS and VSD. Both StableSD and SteinDreamer propose to use control variates to minimize the noisiness of the guidance. Please note, that in both of these works, the excessive variance is compensated by subtracting another noise term by finding coefficients that allow for a better correlation.
>
>
> In our work, however, we explain the second term in SDS not as a control variates, but rather as a projection of the previous step in a re-parametrized DDIM trajectory. Moreover, our variance reduction comes not from using control variates, but from finding a better noise to add to the rendering in the first place. We show that the excessive variance in $\epsilon\_\theta^t(x_t)$ comes from the fact that $x_t$ was obtained using a randomly sampled noise, while our analysis reveals that it should follow a very particular structure (eq.8). By finding a better approximation of the desired noise term we eliminate the root cause of the excessive variance instead of compensating for it later.
>
>
> Unfortunately, the official implementation of Stable Score Distillation is not yet publicly available (nor for SteinDreamer) and we were not able to reproduce their reported results. We will be happy to augment our Table 1 with a quantitative comparison with StableSD as soon as an official implementation gets published.
>
> [1] Ron Mokady et al. “Null-text inversion for editing real images using guided diffusion models”. In: Proceedings of the IEEE/CVF Conference on Computer Vision and Pattern Recognition. 2023, pp. 6038–6047.
>
> [2] Daiki Miyake et al. “Negative-prompt inversion: Fast image inversion for editing with text guided diffusion models”. In: arXiv:2305.16807 (2023).
>
> [3] Wang, Zhengyi, et al. "Prolificdreamer: High-fidelity and diverse text-to-3d generation with variational score distillation." Advances in Neural Information Processing Systems 36 (2024).
>
> [4] Peihao Wang et al. “Steindreamer: Variance reduction for text-to-3d score distillation via stein identity”. In: arXiv:2401.00604 (2023).

---

> > ### Author Response · Authors · 2024-08-11
> >
> > Dear reviewer ZvgL,
> > Thank you again for the recommendations! Could you please let us know if we fully addressed the comments in the review and/or if there is any additional information we could provide? Please note that the author-reviewer discussion period will be over on Tuesday at 23:59 AoE and we won't be able to further reply after this point.
> >
> > Thank you in advance!
> > On behalf of all authors

---

> > > ### Comment · Reviewer_ZvgL · 2024-08-12
> > >
> > > Thanks for the authors' detailed response and explanation! My questions have been largely addressed and I also learned a lot. I hope the authors can summarize these points into the final version. I have increased my score accordingly.

---

> > > > ### Author Response · Authors · 2024-08-12
> > > >
> > > > Thank you for the high evaluation of our work! We are glad that we were able to address the comments and will include a compilation of our answers into the final version.

---

### Official Review · Reviewer_4KTy · 2024-07-14

**Soundness:** 3
**Presentation:** 3
**Contribution:** 3
**Rating:** 6
**Confidence:** 4

**Summary:**

The paper introduces an effective SDS modification that replaces random noise samples in the SDS objective with those obtained with DDIM inversion. The proposed technique enhances text-to-3D generation, outperforming SDS and being comparable to more sophisticated methods.

**Strengths:**

* The paper is well organized and provides valuable intuitions, illustrations, and discussions over reading. The additional discussion in Appendix E is also beneficial.

* The proposed method is simple, reasonable and well motivated. The derivation is clear.

* The experimental results are promising, and the ablation study in Sec 6.2 is highly valuable and interesting.

**Weaknesses:**

* Inversion with negative guidance is surprising, and its rationale remains somewhat unclear. A more detailed investigation and justification would be useful.

* The idea of using DDIM inversion is very similar to ISM[1]. The primary distinction is guided vs unguided inversion. Although the authors justify the guided inversion with negative guidance in Fig. 10, a thorough discussion and detailed comparisons are essential.

* The quantitative evaluation relies on CLIP-IQA, which does not seem quite suitable for 3D generation evaluation. Conducting a user study would be highly valuable. If it is unavailable, some recent works [2,3] aim to propose automated evaluation metrics. For example, would it be reasonable to evaluate ImageReward[4] following [2]?

* The quantitative results miss NFSD and the qualitative comparisons are presented only for two prompts. Could the authors add NFSD to Tab.1 and provide more visual comparisons with all methods?

* There is no evaluation of diversity under a given prompt. It would be useful to include a few generated results for different seeds.

To sum up, I am generally optimistic about the submission but have some concerns regarding the evaluation and close connection with [1], as discussed above.

[1] Liang et al. LucidDreamer: Towards High-Fidelity Text-to-3D Generation via Interval Score Matching.

[2] He et al. T3Bench: Benchmarking Current Progress in Text-to-3D Generation.

[3] Wu et al. GPT-4V(ision) is a Human-Aligned Evaluator for Text-to-3D Generation.

[4] Xu et al. ImageReward: Learning and Evaluating Human Preferences for Text-to-Image Generation.

**Questions:**

* Please address the questions and concerns raised in "Weaknesses".

* In Fig.5, the intermediate $x_t$ do not have noise in the background. DDIM inversion must follow the same marginal distributions as the forward process, but the presented $x_t$ do not. We observed the similar effect for images with monotonic regions if inversion is applied directly to a clean image. We believe such images are likely OOD for the pretrained DM, and injecting slight noise with $\sigma_{min}$ to an image fixed the problem. Do the authors have any thoughts on this? Would the solution above work for SDI and how does it affect its performance?

**Limitations:**

The authors addressed the limitations and potential negative social impact.

---

> ### Author Rebuttal · Authors · 2024-08-07
>
> Thank you for the thoughtful and constructive review. Below we address the points raised in the review:
> # Inversion with negative CFG
> Thank you for pointing this out. We agree that adding this intuition to the main paper will benefit the clarity. Below we support the intuition described in the general response with theoretical and empirical analysis.
>
> ## Experimental analysis of negative CFG
> In the general responce, we base our intuition on the fact that negative CFG attracts generation to the mean of the training dataset. To demonstrate this, we perform a simple experiment of 2D generation with DDIM for different prompts and CFG values of +/-7.5 for each (Fig.3 of the rebuttal PDF). Instead of generating images with opposite prompts, negative CFG directs images to the same class (real estate). This happens because the “real estate” happen to be in the middle of the embedding space, where the process is attracted to.
> ## Theoretical justification
> The effect of attraction to the mean can also be shown more formally. Let’s denote:
>
> $X-\text{full dataset},X_y-\text{prompt conditioned dataset}$
>
> From the optimal denoiser perspective and notations from [1]:
>
> $\hat{\epsilon}\_\theta^t(x_t,\varnothing)=\sum_{i\in X}w_i(x_t)x_i$
>
> $\hat{\epsilon}\_\theta^t(x_t,y)=\sum_{j\in X_y}w_j(x_t)x_j$
>
> Now, rewriting the CFG definition:
>
> $\epsilon_\theta^t(x_t,y)=(1-\gamma)\sum_{i\in X}w_i(x_t)x_i+\gamma \sum_{j\in X_y}w_j(x_t)x_j=$
>
> $=(1-\gamma)[\sum_{i\in X/X_y}w_i(x_t)x_i+\sum_{k\in X_y}w_k(x_t)x_k] + \sum_{j\in X_y}w_j(x_t)x_j=$
>
> $=(1-\gamma)\sum_{i\in X/X_y}w_i(x_t)x_i+\sum_{j\in X_y}w_j(x_t)x_j$
>
>
> In the final formula, image generation is attracted to the images in the dataset relevant to the prompt (independently from CFG), and also is repulsed or attracted to the mean of the rest of the dataset, depending on the sign of the CFG. Consequently, **negative CFG values attract the image to the mean of the dataset and repulse it from the mean on DDIM inversion**.
>
> As mentioned in the general response, our intuition is that *the repulsion from the mean of the dataset in the case of negative CFG serves as a regularization for the noise term and allows to compensate for the bad initialization*.
> # Comparison with ISM
>  Please see the general response above for a detailed comparison between our algorithm and ISM.
> # Additional Metrics
> Thank you for the useful suggestion. We agree that reporting a metric that reflects human preference will make the paper stronger. We adopted the suggested ImageReward. We are using the official implementation and report the rewards directly. Following the same protocol as in Table 1 of the main paper we average the metric across 50 views for each of the 43 prompts. We report the obtained metrics below and plan to augment Table 1 with it.
> |Method|ImageReward ↑|
> |-|-|
> |SDS, 10k steps|-1.51±0.83|
> |SJC, 10k steps|-1.76±0.51|
> |VSD, 25k steps|-1.17±0.58|
> |ESD, 25k steps|-1.20±0.64|
> |HIFA, 25k steps|**-1.16±0.69**|
> |SDI, 10k steps|-1.18±0.59|
> # Additional comparisons
> ## NFSD
> While we compare our method with NFSD qualitatively, we omit quantitative comparison. The official implementation of NFSD is not yet available and we were not able to reproduce its results. We will be happy to add the numerical comparison to Table 1 as soon as an official implementation becomes publicly available.
> ## Additional qualitative comparisons
> In response to more qualitative comparisons apart from the two prompts in the main paper, we would like to kindly point out the numerous qualitative comparisons in the appendix. For this, we follow a protocol adopted across the field: for each baseline, we provide renderings reported in the corresponding papers. To the best of our knowledge, we have reported all prompts and images reported in the corresponding baselines. If considered necessary, we will add visual comparisons on all baselines, using open-source implementations.
> # Evaluation of diversity
> We agree that additional evaluation of diversity will enhance our paper. Please find examples of generations for different seeds and prompts with our method in Fig.4 of the rebuttal PDF. Our generations exhibit a certain degree of diversity but are mostly the same at a coarse level. We did not notice lower diversity compared to SDS[2] or VSD[3].
>
> Additionally, Fig.6 of the rebuttal PDF illustrates interesting behavior, where different human-related prompts generate similar faces. We are not sure if this is a limitation of the underlying diffusion model or our distillation algorithm and plan to address this in future work.
>
> # Noise in the background of x_t
> We agree with the intuition about initial images being out-of-distribution. Indeed, it is rare to find perfectly monotonic backgrounds in natural images. It is unclear if the intermediate $x_t$ is unlikely under the marginal distributions or $x_t$ is close to a mode (in the same way as a completely uniform image has the highest probability density in the i.i.d Gaussian).
>
> In our experiments, however, we did not notice any issues caused by this behavior. Moreover, as we mention in Appendix A.2, we adopt a technique from [4] to fight the Janus problem: we increase the entropy of the process by mixing a small amount of noise during inversion. This helps increase the flexibility of the model to generate non-frontal views. At the same time, it removes the monotonic regions in the noise samples.
>
> [1] Permenter, Frank, and Chenyang Yuan. "Interpreting and Improving Diffusion Models from an Optimization Perspective." arXiv preprint arXiv:2306.04848 (2023).
> [2] Poole, Ben, et al. "Dreamfusion: Text-to-3d using 2d diffusion." arXiv preprint arXiv:2209.14988 (2022).
>
> [3] Wang, Zhengyi, et al. "Prolificdreamer: High-fidelity and diverse text-to-3d generation with variational score distillation." Advances in Neural Information Processing Systems 36 (2024).
>
> [4] Peihao Wang et al. “Taming Mode Collapse in Score Distillation for Text-to-3D Generation”. In: arXiv:2401.00909 (2023).

---

> > ### Author Response · Authors · 2024-08-11
> >
> > Dear reviewer 4KTy,
> >
> > Given the limited time for the author-reviewer discussion period, we would appreciate if you could let us know if we fully addressed your comment and if there are any additional questions or clarifications we could add.
> >
> > Thank you again for your time and effort.
> >
> > Best regards,
> > On behalf of all authors

---

> > > ### Comment · Reviewer_4KTy · 2024-08-11
> > >
> > > I would like to thank the authors for their extensive clarifications, discussions and additional evaluations. My questions have been largely addressed. Overall, I feel more confident about the submission and have increased my score accordingly.

---

> > > > ### Author Response · Authors · 2024-08-12
> > > >
> > > > Thank you, we appreciate your input and the acknowledgment of our work!

---

### Official Review · Reviewer_r6QS · 2024-07-18

**Soundness:** 4
**Presentation:** 4
**Contribution:** 3
**Rating:** 6
**Confidence:** 4

**Summary:**

This paper connects score distillation sampling (SDS) to a DDIM sampling process. The proposed method Score Distillation via Inversion (SDI) replaces the original random noise in SDS with DDIM inversion, and show significantly improved quality compared to SDS and other state-of-the-art prior methods.

**Strengths:**

1. The connection from SDS to DDIM sampling is interesting. The motivation is also clear that it considers the change of the "sample prediction" variable to guide the 3D generation process. The method is supported by well derived theory (for 2D) and the DDIM inversion is a clever solution.
2. The results generally look good. It shows significant improvement over its baseline method SDS, and is competitive to recent state-of-the-art text-to-3D methods. The experiments are thorough.
3. The proposed method and results could provide insights for future research in the important direction of text-to-3D distillation.

**Weaknesses:**

1. Overall, the visual results seem to be having a little bit "gray" color/style shift. Is this due to the approximation or some bias in the theory? Is there any hypothesis for theoretical / practical reasons?
2. The ddim inversion and approximation could introduce additional computation cost and training noise.
3. In "ISM as a special case", it seems the main difference of the proposed method and ISM is having the additional text condition. Is there any other key differences in method implementation or theory? More details about the contribution of this work based on ISM could be helpful.

**Questions:**

Does the proposed method remedy Janus problem, compared to the baseline SDS method? (If yes, why)
Also please check the weakness section.

**Limitations:**

The authors discussed limitations.

---

> ### Author Rebuttal · Authors · 2024-08-07
>
> We thank r6QS for the helpful review. We will address their comments in the final version and clarify some points below:
> # Gray color shift
> We agree that results in the main paper might seem gray or sometimes have a red/green tone. We explain this by the dark background color that is generated in the NeRF. Our algorithm is implemented on top of threestudio’s SDS, where apart from the object-level NeRF, the shapes are also allowed to have a background. Typically the generation converges to darker tones of the background, which biases the diffusion process to reflect illumination in the rendered 3D shapes. For visualization in the original submission, we simply cropped the object from the final renderings. When on a white background, this removes the illumination context and makes the shapes look grayer.
>
>
>
> To further illustrate this we provide renderings with their original background in Fig.7 of the rebuttal PDF. Additionally, Fig. 1 (bottom row) of the rebuttal PDF shows that when our algorithm is reimplemented in the ISM’s code base, where the background is fixed to be white, our final renderings have brighter, white colors reflecting the global illumination.
>
> In summary, **the diffusion model bakes in the background illumination into the texture of the shape.**
> One possible solution that was suggested in prior work consists of randomly changing the background colors to avoid overfitting to a single one. In this case, however, we observed a decrease in the quality of the renderings.
>
>
> # Computational Cost
> We agree that, unfortunately, DDIM inversion requires additional forward passes with the denoiser, which increases computational cost. However, we would like to clarify a few points:
>
> 1. Despite the additional cost required for obtaining the noise sample, our algorithm is typically more stable and requires fewer steps to converge than other state-of-the-art methods, all while having similar quality in the results. The time per generation in Table 1 shows that **our algorithm is still 2-3x times faster than state-of-the-art** since it takes fewer steps.
>
> 2. Our ablation in Fig.11 demonstrates that 10 steps are enough for DDIM inversion to obtain good-quality 3D generations. In practice, one NeRF update is so slow that after the 10 additional inferences with the denoiser per optimization step we see only a 2x slow-down. This effect will be more noticeable in future work involving Gaussian Splatting, where 3D shape update is not a bottleneck anymore.
> # Differences with ISM
> For a detailed comparison of our algorithm and ISM, please refer to the general response above and to the rebuttal PDF attached to the response.
>
> # Janus problem
> Thank you for the question. Without additional regularization, our algorithm has a stronger tendency towards the Janus problem compared to SDS. SDS uses prompt augmentation to avoid this problem, adding phrases like “front view” and “back view.” As in Appendix A2, our algorithm’s stronger bias toward the Janus problem can be explained by the property of the diffusion models to ignore parts of the prompt when CFG is not high enough (see Fig.3 in the main paper). The proposed improvements of finding a better noise term allow us to reduce the CFG values from 100 in SDS to the typical 7.5, which leads to a weaker view-dependent prompt augmentation.
>
> To additionally prevent the Janus problem, we add entropy as in [1], and orthogonalize the augmented prompts as in [2]. In rare cases, our algorithm still might produce multiple faces in one generation. This limitation is reported and illustrated in the appendix E1.
>
> [1] Peihao Wang et al. “Taming Mode Collapse in Score Distillation for Text-to-3D Generation”. In: arXiv:2401.00909 (2023).
>
> [2] Mohammadreza Armandpour et al. “Re-imagine the negative prompt algorithm: Transform 2d diffusion into 3d, alleviate Janus problem and beyond”. In: arXiv:2304.04968 (2023).

---

> > ### Author Response · Authors · 2024-08-11
> >
> > Dear reviewer r6QS,
> >
> > Thank you again for the thorough review and high evaluation of our work. Could you please let us know if our response fully addressed your comments and if we can add any additional clarifications ? Please note the additional evaluations we added in the PDF attached to the general response.
> >
> > Best regards,
> > On behalf of all authors

---

### Author Rebuttal · Authors · 2024-08-07

We thank the reviewers for the detailed and thoughtful feedback.

We are pleased that they appreciate the theoretical contributions and the novelty of our approach (“*well organized and provides valuable intuitions*” (4KTy), “*insights for future research in the important direction*” (r6QS)). The reviewers found our method to be practical (“*significantly improves the quality of 3D shape generation*” (ZvgL)) and well supported by experiments (“*highly valuable and interesting ablation study*” (4KTy), “*experiments are thorough*” (r6QS), “*thorough analysis and convincing ablation*” (ZvgL)).

In response to requests for further clarifications and comparisons, we were able to execute most of the suggested experiments and plan to add them to the final version of the paper. **Please refer to the rebuttal PDF.**

Below, we address common questions or questions providing valuable intuition, while the rest of the comments are addressed in individual replies.
# Comparison with ISM
We agree with reviewers r6QS and 4KTy that Interval Score Matching (ISM) is relevant since it uses DDIM inversion to improve the consistency of 3D guidance. Below we highlight the key differences between our work and ISM.
## Theoretical assumptions
ISM empirically observes that ``pseudo-GT’’ images used to guide Score Distillation Sampling (SDS) are sensitive to their input and that the single-step generation of pseudo-GT yields oversmoothing. From these observations, **starting with SDS guidance**, ISM adds DDIM inversion and multi-step generation to **empirically** improve the stability of the guidance.

As highlighted by the reviewer ZvgL, most advances in score distillation—including ISM—were obtained experimentally. In contrast, our work **starts with 2D diffusion sampling** to rederive score-distillation guidance and motivate improvements. That is, our work formally connects SDS to well-justified techniques in 2D sampling.
## DDIM inversion
In ISM the empirically motivated **DDIM inversion is at the basis of the derivation** of the final update rule. We suggest a general form of the noise term (eq.8), for which **DDIM inversion is just one possible solution**. Our theoretical insights are agnostic to particular algorithms of root-finding, which makes it possible to use more efficient solutions in future research (e.g. train diffusion models as invertible maps to sample noise faster).
## Guidance term
The update rules provided in our eq.10 and ISM’s eq.17 have two main differences:

1. **Full vs. interval noise.** Assuming DDIM inversion finds a perfect root of our stationary equation, ISM’s update rule can take a similar form to ours eq.10 (interval noise is equal to $\kappa$ if eq.8 is satisfied). However, as shown in Fig.9, DDIM inversion does not find a perfect root, and thus the two forms are not equivalent. Our theory shows that the **full noise term is more accurate**. We also show the effect of choosing one term vs. another in Fig.2 of the rebuttal PDF.

2. **Conditional vs. unconditional inversion.** Our derivation hints that the roots of eq.8 are prompt-dependent, motivating our use of conditional DDIM inversion (not used in ISM). Our Fig.10 shows how unconditional inversion yields oversaturation. To demonstrate this effect more clearly, Fig.5 of the rebuttal PDF provides a simple 2D experiment.

## Practical Results
To control for different design choices (GaussianSplattings in ISM vs. NeRF in ours, etc.) **we reimplemented our algorithm in the code base of ISM**, with only the minimal changes discussed in the previous section.

Fig.1 of the rebuttal PDF provides a qualitative comparison using the prompts and settings in ISM’s code. Fig.2 shows the effect of each change made to ISM guidance. Below is the quantitative comparison (ISM code base for both).
|Method|CLIP Score ↑|quality ↑|sharpness ↑|real ↑|ImageReward ↑|Time|VRAM|
|-|-|-|-|-|-|-|-|
|ISM, 5k steps|**28.60±2.03**|0.85±0.02|0.98±0.01|0.98±0.01|-0.52±0.48|45min|15.4GB|
|SDI (ours), 5k steps|28.47±1.29|**0.88±0.03**|**0.99±0.00**|0.98±0.01|**-0.30±0.32**|43min|15.4GB|

## Summary
We bridge the gap between experimentally-based score distillation techniques and theoretically-justified 2d sampling. Both SDS and ISM can be seen as different approaches to finding roots of eq.8., This theoretical insight allows us to modify both ISM and SDS, reducing over-saturation for the first and improving general quality of the second.


# Smaller error in eq.8 = better 3D generation?
In response to reviewer ZvgL, we clarify the intuition behind the relation between 3D generation quality and error in root-finding of the stationary point equation.
A more accurate solution does not necessarily yield a better 3D generation. It is better to say that **a more precise solution of eq.8 makes the guidance of score distillation closer to that of DDIM**. However, other factors contribute to the quality of 3D shape: initialization, view consistency, denoiser’s generative ability, etc. We address this question in more detail in the individual reply to reviewer ZvgL.
# Why negative CFG?
In response to reviewer 4KTy, we add intuition behind using negative CFG. While the general noise term is formally derived, the choice of negative CFG is mostly intuition-based and supported by experiments.

Contrary to a possible impression, negative CFG is not equal to a prompt of an opposite meaning. In fact, **negative CFG values attract the image to the mean of the dataset in the generation process and repulse it from the mean on DDIM inversion.** To support this claim we provide a theoretical and empirical argument in the individual response to reviewer 4KTy.

We notice that the NeRF initialization in SDS is very different from the i.i.d. Gaussian initialization in DDIM. Our intuition is that *negative CFG acts as regularization of the noise due to its mean-repulsion properties*. It brings the bad initialization of the NeRF closer to the Gaussian noise expected by DDIM.

---

### Decision · Program_Chairs · 2024-09-25

**Decision:**

Accept (poster)

**Comment:**

The paper introduces Score Distillation via Inversion (SDI), a method that enhances 3D shape generation by improving the Score Distillation Sampling (SDS) process. SDI connects SDS with Denoising Diffusion Implicit Models (DDIM), reducing over-smoothing and preserving detail in 3D models. This approach improves 3D generation quality to match that of 2D images without needing additional training or multi-view supervision. SDI connects SDS with DDIM, providing deeper insights into the discrepancies between 2D and 3D generation. The method improves 3D generation quality without the need for additional training or complex setups. After carefully reading the paper, reviews, and rebuttals, the AC recommends to accept the paper.